# Foundations of Comparison-Based Hierarchical Clustering

**Debarghya Ghoshdastidar**[*][†]
Department of Informatics, TU Munich
ghoshdas@in.tum.de

**Michaël Perrot**[*]
Max Planck Institute for Intelligent Systems
michael.perrot@tuebingen.mpg.de

**Ulrike von Luxburg**
Department of Computer Science, University of Tübingen
Max Planck Institute for Intelligent Systems
luxburg@informatik.uni-tuebingen.de

## Abstract

We address the classical problem of hierarchical clustering, but in a framework where one does not have access to a representation of the objects or their pairwise similarities. Instead, we assume that only a set of comparisons between objects is available, that is, statements of the form "objects $i$ and $j$ are more similar than objects $k$ and $l$." Such a scenario is commonly encountered in crowdsourcing applications. The focus of this work is to develop comparison-based hierarchical clustering algorithms that do not rely on the principles of ordinal embedding. We show that single and complete linkage are inherently comparison-based and we develop variants of average linkage. We provide statistical guarantees for the different methods under a planted hierarchical partition model. We also empirically demonstrate the performance of the proposed approaches on several datasets.

## 1 Introduction

The definition of clustering as *the task of grouping similar objects* emphasizes the importance of assessing similarity scores for the process of clustering. Unfortunately, many applications of data analysis, particularly in crowdsourcing and psychometric problems, do not come with a natural representation of the underlying objects or a well-defined similarity function between pairs of objects. Instead, one only has access to the results of comparisons of similarities, for instance, quadruplet comparisons of the form "similarity between $x_i$ and $x_j$ is larger than similarity between $x_k$ and $x_l$."

The importance and robustness of collecting such ordinal information from human subjects and crowds has been widely discussed in the psychometric and crowdsourcing literature (Shepard, 1962; Young, 1987; Borg and Groenen, 2005; Stewart et al., 2005). Subsequently, there has been growing interest in the machine learning and statistics communities to perform data analysis in a comparison-based framework (Agarwal et al., 2007; Van Der Maaten and Weinberger, 2012; Heikinheimo and Ukkonen, 2013; Zhang et al., 2015; Arias-Castro et al., 2017; Haghiri et al., 2018). The traditional approach for learning in an ordinal setup involves a two step procedure—first obtain a Euclidean embedding of the objects from available similarity comparisons, and subsequently learn from the embedded data using standard machine learning techniques (Borg and Groenen, 2005; Agarwal et al., 2007; Jamieson and Nowak, 2011; Tamuz et al., 2011; Van Der Maaten and Weinberger, 2012; Terada and von Luxburg, 2014; Amid and Ukkonen, 2015). As a consequence, the statistical

---

[*]Both authors contributed equally to the paper.
[†]This work was done when the author was affiliated to the University of Tübingen.

performance of the resulting comparison-based learning algorithms relies both on the goodness of the embedding and the subsequent statistical consistency of learning from the embedded data. While there exists theoretical guarantees on the accuracy of ordinal embedding (Jamieson and Nowak, 2011; Kleindessner and Luxburg, 2014; Jain et al., 2016; Arias-Castro et al., 2017), it is not known if one can design provably consistent learning algorithms using mutually dependent embedded data points.

An alternative approach, which has become popular in recent years, is to directly learn from the ordinal relations. This approach has been used for estimation of data dimension, centroid or density (Kleindessner and Luxburg, 2015; Heikinheimo and Ukkonen, 2013; Ukkonen et al., 2015), object retrieval and nearest neighbour search (Kazemi et al., 2018; Haghiri et al., 2017), classification and regression (Haghiri et al., 2018), clustering (Kleindessner and von Luxburg, 2017a; Ukkonen, 2017), as well as hierarchical clustering (Vikram and Dasgupta, 2016; Emamjomeh-Zadeh and Kempe, 2018). The theoretical advantage of a direct learning principle over an indirect embedding-based approach is reflected by the fact that some of the above works come with statistical guarantees for learning from ordinal comparisons (Haghiri et al., 2017, 2018; Kazemi et al., 2018).

**Motivation.**   The motivation for the present work arises from the absence of comparison-based clustering algorithms that have strong statistical guarantees, or more generally, the limited theory in the context of comparison-based clustering and hierarchical clustering. While theoretical foundations of standard hierarchical clustering can be found in the literature (Hartigan, 1981; Chaudhuri et al., 2014; Dasgupta, 2016; Moseley and Wang, 2017), corresponding works in the ordinal setup has been limited (Emamjomeh-Zadeh and Kempe, 2018). A naive approach to derive guarantees for comparison-based clustering would be to combine the analysis of a classic clustering or hierarchical clustering algorithm with existing guarantees for ordinal embedding (Arias-Castro et al., 2017). Unfortunately, this does not work since the known worst-case error rates for ordinal embedding are too weak to provide any reasonable guarantee for the resulting comparison-based clustering algorithm. The existing guarantees for ordinal hierarchical clustering hold under a triplet framework, where each comparison returns the two most similar among three objects (Emamjomeh-Zadeh and Kempe, 2018). The results show that the underlying hierarchy can be recovered by few adaptively chosen comparisons, but if the comparisons are provided beforehand, which is the case in crowdsourcing, then the number of required comparisons is rather large. The focus of the present work is to develop provable comparison-based hierarchical clustering algorithms that can find an underlying hierarchy in a set of objects given either adaptively or non-adaptively chosen sets of comparisons.

**Contribution 1: Agglomerative algorithms for comparison-based clustering.**   The only known hierarchical clustering algorithm in a comparison-based framework employs a divisive approach (Emamjomeh-Zadeh and Kempe, 2018). We observe that it is easy to perform agglomerative hierarchical clustering using only comparisons since one can directly reformulate single linkage and complete linkage clustering algorithms in the quadruplet comparisons framework. However, it is well known that single and complete linkage algorithms typically have poor worst-case guarantees (Cohen-Addad et al., 2018). While average linkage clustering has stronger theoretical guarantees (Moseley and Wang, 2017; Cohen-Addad et al., 2018), it cannot be used in the comparison-based setup since it relies on an averaging of similarity scores. We propose two variants of average linkage clustering that can be applied to the quadruplet comparisons framework. We numerically compare the merits of these new methods with single and complete linkage and embedding based approaches.

**Contribution 2: Guarantees for true hierarchy recovery.**   Dasgupta (2016) provided a new perspective for hierarchical clustering in terms of optimizing a cost function that depends on the pairwise similarities between objects. Subsequently, theoretical research has focused on worst-case analysis of different algorithms with respect to this cost function (Roy and Pokutta, 2016; Moseley and Wang, 2017; Cohen-Addad et al., 2018). However, such an analysis is complicated in an ordinal setup, where the algorithm is oblivious to the pairwise similarities. In this case, one can study a stronger notion of guarantee in terms of exact recovery of the true hierarchy (Emamjomeh-Zadeh and Kempe, 2018). That work, however, considers a simplistic noise model, where the result of each comparison may be randomly flipped independently of other comparisons (Jain et al., 2016). Such an independent noise can be easily tackled by repeatedly querying the same comparison and using a majority vote. It cannot account for noise in the underlying objects and their associated similarities. Instead, we consider a theoretical model that generates random pairwise similarities with a planted hierarchical structure (Balakrishnan et al., 2011). This induces considerable dependence among

**input**  : Set of objects $\mathcal{X} = \{x_1, \ldots, x_N\}$; Cluster-level similarity $W : 2^{\mathcal{X}} \times 2^{\mathcal{X}} \to \mathbb{R}$.
**output**: Binary tree, or dendrogram, representing a hierarchical clustering of $\mathcal{X}$.
**begin**
&emsp;| Let $\mathcal{B}$ be a collection of $N$ singleton trees $\mathcal{C}_1, \ldots, \mathcal{C}_N$ with root nodes $\mathcal{C}_i.root = \{x_i\}$.
&emsp;| **while** $|\mathcal{B}| > 1$ **do**
&emsp;&emsp;| Let $\mathcal{C}, \mathcal{C}'$ be the pair of trees in $\mathcal{B}$ for which $W(\mathcal{C}.root, \mathcal{C}'.root)$ is maximum.
&emsp;&emsp;| Create $\mathcal{C}''$ with $\mathcal{C}''.root = \{\mathcal{C}.root \cup \mathcal{C}'.root\}$, $\mathcal{C}''.left = \mathcal{C}$, and $\mathcal{C}''.right = \mathcal{C}'$.
&emsp;&emsp;| Add $\mathcal{C}''$ to the collection $\mathcal{B}$, and remove $\mathcal{C}, \mathcal{C}'$.
&emsp;| **end**
&emsp;| **return** The surviving element in $\mathcal{B}$.
**end**

**Algorithm 1:** Agglomerative Hierarchical Clustering.

the quadruplets, and makes the analysis challenging. We derive conditions under which different comparison-based agglomerative algorithms can exactly recover the hierarchy with high probability.

## 2   Background

In this section we introduce standard hierarchical clustering with known similarities, we describe the model used for the theoretical analyses, and we formalize the comparison-based framework.

### 2.1   Agglomerative hierarchical clustering with known similarity scores

Let $\mathcal{X} = \{x_i\}_{i=1}^N$ be a set of $N$ objects, which may not have a known feature representation. We assume that there exists an underlying symmetric similarity function $w : \mathcal{X} \times \mathcal{X} \to \mathbb{R}$. The goal of hierarchical clustering is to group the $N$ objects to form a binary tree such that $x_i$ and $x_j$ are merged in the bottom of the tree if their similarity score $w_{ij} = w(x_i, x_j)$ is high, and vice-versa. Here, we briefly review popular agglomerative clustering algorithms (Cohen-Addad et al., 2018). They rely on the similarity score $w$ between objects to define a similarity function between any two clusters, $W : 2^{\mathcal{X}} \times 2^{\mathcal{X}} \to \mathbb{R}$. Starting from $N$ singleton clusters, each iteration of the algorithm merges the two most similar clusters. This is described in Algorithm 1, where different choices of $W$ lead to different algorithms. Given two clusters $G$ and $G'$, popular choices for $W(G, G')$ are

$$W(G, G') = \underbrace{\max_{x_i \in G, x_j \in G'} w_{ij}}_{\textbf{Single Linkage (SL)}}, \quad \text{or} \quad \underbrace{\min_{x_i \in G, x_j \in G'} w_{ij}}_{\textbf{Complete Linkage (CL)}}, \quad \text{or} \quad \underbrace{\sum_{x_i \in G, x_j \in G'} \frac{w_{ij}}{|G||G'|}}_{\textbf{Average Linkage (AL)}}.$$

### 2.2   Planted hierarchical model

Theoretically, we study the problem of hierarchical clustering under a *noisy hierarchical block matrix* (Balakrishnan et al., 2011) where, given $N$ objects, the matrix of pairwise similarities can be written as $M + R$, where $M = (\mu_{ij})_{1 \le i,j \le N}$ is a symmetric ideal similarity matrix characterizing the planted hierarchy among the examples and $R = (r_{ij})_{1 \le i,j \le N}$ is a symmetric perturbation matrix that accounts for the noise in the observed similarity scores. In this paper, we assume that the entries $\{r_{ij}\}_{1 \le i < j \le N}$ are mutually independent and normally distributed, that is $r_{ij} \sim \mathcal{N}\left(0, \sigma^2\right)$, for some fixed variance $\sigma^2$. The ideal similarity matrix $M$ is constructed in the following way. We assume that the planted hierarchy is a balanced binary tree of height $L$ (see Figure 1), where the $2^L$ leaf nodes $\mathcal{G}_1, \ldots, \mathcal{G}_{2^L}$ correspond to "pure clusters", each of size $N_0$. Thus, the total number of objects in $\mathcal{X}$ is $N = N_0 2^L$. For some constants $\delta > 0$ and $\mu$, the ideal similarities are defined as follows:
**Step-0:** $\mathcal{X}$ is divided into two equal sized clusters, and, given $x_i$ and $x_j$ lying in different clusters, their ideal similarity is set to $\mu_{ij} = \mu - L\delta$ (dark blue off-diagonal block in Figure 1).
**Step-1:** Each of the two groups is further divided into two sub-groups, and, for each pair $x_i, x_j$ separated due to this sub-group formation, we set $\mu_{ij} = \mu - (L-1)\delta$.
**Step-2,\ldots, $L-1$:** The above process is repeated $L - 1$ times, and in step $\ell$, the ideal similarity across two newly-formed sub-groups is $\mu_{ij} = \mu - (L - \ell)\delta$.

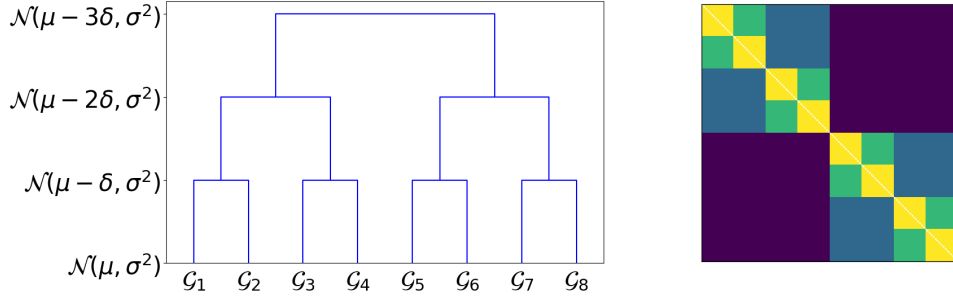

Figure 1: **(Left)** Illustration of the planted hierarchical model for $L = 3$ along with specification of the distributions for similarities at different levels; **(Right)** Hierarchical block structure in the expected pairwise similarity matrix, where darker implies smaller similarity.

**Step-$L$:** The above steps form $2^L$ clusters, $\mathcal{G}_1, \ldots, \mathcal{G}_{2^L}$, each of size $N_0$. The ideal similarity between two objects $x_i, x_j$ belonging to the same cluster is $\mu_{ij} = \mu$ (yellow blocks in Figure 1).

This gives rise to similarities of the form $w_{ij} = \mu_{ij} + r_{ij}$ for all $i < j$. By symmetry of $M$ and $R$, $w_{ji} = w_{ij}$. We can equivalently assume that, for all $i < j$, the similarities are independently drawn as $w_{ij} = w_{ji} \sim \mathcal{N}\left(\mu_{ij}, \sigma^2\right)$. Note that the pairwise similarity gets smaller in expectation when two objects are merged higher in the true hierarchy. We consider the problem of exact recovery of the above planted structure, that is correct identification of all the pure clusters $\mathcal{G}_1, \ldots, \mathcal{G}_{2^L}$ and recovery of the entire hierarchy among the clusters.

### 2.3 The comparison-based framework

In Section 2.1 we assumed that, even without a representation of the objects, we had access to a similarity function $w$. In the rest of this paper, we consider the ordinal setting, where $w$ is not available, and information about similarities can only be accessed through quadruplet comparisons. We assume that we are given a set $\mathcal{Q} \subseteq \{(i, j, k, l) : x_i, x_j, x_k, x_l \in \mathcal{X}, w_{ij} > w_{kl}\}$, that is, for every ordered tuple $(i, j, k, l) \in \mathcal{Q}$, we know that $x_i$ and $x_j$ are more similar than $x_k$ and $x_l$. There exists a total of $\mathcal{O}\left(N^4\right)$ quadruplets, but in a practical crowdsourcing application, the available set $\mathcal{Q}$ may only be a small subset of all possible quadruplets. Since noise is inherent in the similarities, we do not consider it in the comparisons. We assume $\mathcal{Q}$ is obtained in either of the two following ways:
**Active comparisons:** In this case, the algorithm can adaptively ask an oracle quadruplet queries of the form $w_{ij} \gtrless w_{kl}$ and the outcome will be either $w_{ij} > w_{kl}$ or $w_{ij} < w_{kl}$.
**Passive comparisons:** In this case, for every tuple $(i, j, k, l)$, we assume that with some sampling probability $p \in (0, 1]$, there is a comparison $w_{ij} \gtrless w_{kl}$ and based on the outcome either $(i, j, k, l) \in \mathcal{Q}$ or $(k, l, i, j) \in \mathcal{Q}$. We also assume that the observations of the quadruplets are independent.

## 3  Comparison-based hierarchical clustering

In this section, we discuss that single linkage and complete linkage can be easily implemented in the comparison-based setting, provided that we have access to $\Omega\left(N^2\right)$ adaptively selected quadruplets. However, their statistical guarantees are very weak. It prompts us to study two variants of average linkage. On the one hand, Quadruplets-based Average Linkage (4–AL) uses average linkage-like ideas to directly estimate the cluster level similarities from the quadruplet comparisons. On the other hand, Quadruplets Kernel Average Linkage (4K–AL) uses the quadruplet comparisons to estimate the similarities between the different objects and then uses standard average linkage. We show that both of these variants have good statistical performances in the following senses: (i) they can exactly recover the planted hierarchy under mild assumptions on the signal-to-noise ratio $\frac{\delta}{\sigma}$ and the size of the pure clusters $N_0 = \frac{N}{2^L}$ in the model introduced in Section 2.2, (ii) 4K–AL only needs $\mathcal{O}\left(N \ln N\right)$ active comparisons to achieve exact recovery, and (iii) both 4K–AL and 4–AL can achieve exact recovery using only a small subset of passively obtained quadruplets (sampling probability $p \ll 1$).

## 3.1 Single linkage (SL) and complete linkage (CL)

The single and complete linkage algorithms inherently fall in the comparison-based framework. To see this, first notice that the $\arg\max$ and $\arg\min$ functions used in these methods only depend on quadruplet comparisons. Although it is not possible to exactly compute the linkage value $W(G, G')$, one can retrieve, in each cluster, the pair of objects that achieve the maximum or minimum similarity. Then, the knowledge of these optimal object pairs is sufficient since our primary aim is to find the pair of clusters $G, G'$ that maximizes $W(G, G')$ and this can be easily achieved through quadruplet comparisons between the optimal object pairs of every $G, G'$. This discussion emphasizes that CL and SL fall well in the comparison-based framework when the quadruplets can be adaptively chosen—in order to select pairs with minimum or maximum similarities. The next proposition, proved in the appendix, bounds the number of active comparisons necessary and sufficient to use SL and CL.

**Proposition 1 (Active query complexity of SL and CL).** *The SL and CL algorithms require at least $\Omega(N^2)$ and at most $\mathcal{O}(N^2 \ln N)$ number of active quadruplet comparisons.*

We now state a sufficient condition for exact recovery of the planted model for both SL and CL as well as a matching (up to constant) necessary condition for SL. The proof is in the appendix.

**Theorem 1 (Exact recovery of planted hierarchy by SL and CL).** *Assume that $\eta \in (0, 1)$. If $\frac{\delta}{\sigma} \geq 4\sqrt{\ln\left(\frac{N}{\eta}\right)}$, then SL and CL exactly recover the planted hierarchy with probability $1 - \eta$. Conversely, for $\frac{\delta}{\sigma} \leq \frac{1}{4}\sqrt{\ln\left(\frac{N}{2^L}\right)}$ and large $\frac{N}{2^L}$, SL fails to recover the hierarchy with probability $\frac{1}{2}$.*

Theorem 1 implies that a necessary and sufficient condition for exact recovery by single linkage is that the signal-to-noise ratio grows as $\sqrt{\ln N}$ with the number of examples. This strong requirement raises the question of whether one can achieve exact recovery under weaker assumptions and with less quadruplets. The subsequent sections provide an affirmative answer to this question.

## 3.2 Quadruplets kernel average linkage (4K–AL)

Average linkage is difficult to cast to the ordinal framework due to the averaging of pairwise similarities, $w_{ij}$, which cannot be computed using only comparisons. A first way to overcome this issue is to use the quadruplet comparisons to derive some kind of proxies for the similarities $w_{ij}$. These proxy similarities can then be directly used in the standard formulation of average linkage. To derive them we use ideas that are close in spirit to the triplet comparisons-based kernel developed by Kleindessner and von Luxburg (2017a). Furthermore, we propose two different definitions depending on whether we use active comparisons (Equation 1) or passive comparisons (Equation 3).

**Active case.** We first consider the active case, where the quadruplet comparisons to be evaluated can be chosen by the algorithm. A pair of distinct items $(i_0, j_0)$ is chosen uniformly at random, and a set of landmark points $\mathcal{S}$ is constructed such that every $k \in \{1, \ldots, N\}$ is independently added to $\mathcal{S}$ with probability $q$. The proxy similarity between two distinct objects $x_i$ and $x_j$ is then defined as

$$K_{ij} = \sum_{k \in \mathcal{S}\setminus\{i,j\}} \left(\mathbb{I}_{(w_{ik} > w_{i_0 j_0})} - \mathbb{I}_{(w_{ik} < w_{i_0 j_0})}\right)\left(\mathbb{I}_{(w_{jk} > w_{i_0 j_0})} - \mathbb{I}_{(w_{jk} < w_{i_0 j_0})}\right). \tag{1}$$

The underlying idea is that two similar objects should behave similarly with respect to any third object, that is if $x_i$ and $x_j$ are similar then we should have $w_{ik} \approx w_{jk}$ for any other object $x_k$. Since we cannot directly access the similarities, we instead use comparisons to a reference similarity $w_{i_0 j_0}$ to evaluate the closeness between $w_{ik}$ and $w_{jk}$.

The next theorem presents exact recovery guarantees for 4K–AL with actively obtained comparisons.

**Theorem 2 (Exact recovery of planted hierarchy by 4K–AL with active comparisons).** *Let $\eta \in (0, 1)$ and $\Delta = \frac{\eta^2}{100}\frac{\delta}{\sigma}e^{-2L^2\delta^2/\sigma^2}$. There exists an absolute constant $C > 0$ such that if $N_0 > \frac{4}{\Delta}\sqrt{N}$ and we set $q > \max\left\{C\frac{2^{2L}}{N\Delta^4}\ln\left(\frac{N}{\eta}\right), \frac{3}{N}\ln\left(\frac{2}{\eta}\right)\right\}$, then with probability at least $1 - \eta$, 4K–AL exactly recovers the planted hierarchy using at most $2qN^2$ number of actively chosen quadruplet comparisons.*

*In particular, if $L = \mathcal{O}(1)$, the above statement implies that even with $\frac{\delta}{\sigma}$ constant, 4K–AL exactly recovers the planted hierarchy with probability $1 - \eta$ using only $\mathcal{O}(N \ln N)$ active comparisons.*

The above result shows that, in comparison to SL or CL, the proposed 4K–AL method achieves consistency for smaller signal-to-noise ratio $\frac{\delta}{\sigma}$, and can also do so with only $\mathcal{O}(N \ln N)$ active comparisons, which is much smaller than that needed by SL and CL. Our result also aligns with the conclusion of Emamjomeh-Zadeh and Kempe (2018), who showed that $\mathcal{O}(N \ln N)$ active triplet comparisons suffice to recover hierarchy under a different (data-independent) noise model. It is also worth noting that the condition $N_0 = \Omega\left(\sqrt{N}\right)$ is necessary for exact recovery of the planted hierarchy since the condition is necessary even in the case of planted flat clustering (Chen and Xu, 2016, Figure 1).

From a theoretical perspective, it is sufficient to use a single random reference similarity $w_{i_0 j_0}$. However, in practice, we observe better performances when considering a set $\mathcal{R}$ of multiple reference pairs. Hence, in the experiments, we use the following extension of the above kernel function:

$$K_{ij} = \sum_{(i_0, j_0) \in \mathcal{R}} \sum_{k \in \mathcal{S} \setminus \{i, j\}} \left( \mathbb{I}_{\left(w_{ik} > w_{i_0 j_0}\right)} - \mathbb{I}_{\left(w_{ik} < w_{i_0 j_0}\right)} \right) \left( \mathbb{I}_{\left(w_{jk} > w_{i_0 j_0}\right)} - \mathbb{I}_{\left(w_{jk} < w_{i_0 j_0}\right)} \right). \quad (2)$$

**Passive case.** Theorem 2 shows that 4K–AL can exactly recover the planted hierarchy even for a constant signal-to-noise ratio, provided that it can actively choose the quadruplets. It is natural to ask if the same holds in the passive case, where we do not have the freedom of querying specific comparisons but instead have access to a small pre-computed set of quadruplet comparisons $\mathcal{Q}$. We address this problem using the following variant of the aforementioned quadruplets kernel:

$$K_{ij} = \sum_{k, l = 1, k < l}^{N} \sum_{r=1}^{N} \left( \mathbb{I}_{(i, r, k, l) \in \mathcal{Q}} - \mathbb{I}_{(k, l, i, r) \in \mathcal{Q}} \right) \left( \mathbb{I}_{(j, r, k, l) \in \mathcal{Q}} - \mathbb{I}_{(k, l, j, r) \in \mathcal{Q}} \right) \quad (3)$$

for all $i \neq j$. This formulation extends the active kernel in (1) by using all $\binom{N}{2}$ pairs of $(k, l)$ as reference similarities instead of a single pair $(i_0, j_0)$. But each term in the sum contributes only when we simultaneously observe the comparisons between $(i, r)$ and $(k, l)$ and between $(j, r)$ and $(k, l)$. Theorem 3 presents guarantees for 4K–AL with quadruplets obtained from the passive comparisons model in Section 2.3.

**Theorem 3 (Exact recovery of planted hierarchy by 4K–AL with passive comparisons).** *Let $\eta \in (0, 1)$ and $\Delta = \frac{\delta}{2\sigma} e^{-L^2 \delta^2 / 4\sigma^2}$. There exists an absolute constant $C > 0$ such that if $N_0 > \frac{8}{\Delta}\sqrt{N}$ and we set $p > \max\left\{ C\frac{2^L}{\Delta^2}\sqrt{\frac{1}{N}\ln\left(\frac{N}{\eta}\right)}, \frac{2}{N^4}\ln\left(\frac{2}{\eta}\right) \right\}$, then with probability at least $1 - \eta$, the 4K–AL algorithm exactly recovers the planted hierarchy using at most $pN^4$ quadruplet comparisons, which are passively obtained based on the model described in Section 2.3.*

*In particular, if $L = \mathcal{O}(1)$, the above statement implies that even with $\frac{\delta}{\sigma}$ constant, 4K–AL exactly recovers the planted hierarchy with probability $1 - \eta$ using $\mathcal{O}\left(N^{7/2} \ln N\right)$ passive comparisons.*

The derived conditions for exact recovery are similar to Theorem 2 in terms of $\frac{\delta}{\sigma}$, but passive 4K–AL requires a much larger number of passive comparisons than active 4K–AL. While this may seem disappointing, $\mathcal{O}\left(N^{7/2}\right)$ passive comparisons might, in fact, be necessary in this case. Indeed, Emamjomeh-Zadeh and Kempe (2018, Theorem 2.3) show that in the case of triplets, $\Omega\left(N^3\right)$ passive triplet comparisons are necessary to exactly recover a hierarchy in the worst case. The proof can be easily adapted to the quadruplet comparison setting to prove a worst-case complexity of $\Omega\left(N^4\right)$ passive quadruplets. The above result shows that under the planted model, which is simpler than the worst-case, the query complexity can be improved at least by a factor of $\sqrt{N}$. Further study is required to identify a precise necessary condition. We also believe that the passive query complexity should reduce considerably if the signal-to-noise ratio $\frac{\delta}{\sigma}$ grows with $N$.

### 3.3 Quadruplets-based average linkage (4–AL)

In 4K–AL we derived a proxy for the similarities between objects and then used standard average linkage. In this section we consider a different approach where we use the quadruplet comparisons to define a new cluster-level similarity function. This method is particularly well suited when it is not possible to actively query the comparisons. We assume that we are given a set of passively

obtained quadruplets $\mathcal{Q}$ as in the previous section (4K–AL with passive comparisons). Using this set of comparisons, one can estimate the relative similarity between two pairs of clusters. For instance, let $G_1, G_2, G_3, G_4$ be four clusters such that $G_1, G_2$ are disjoint and so are $G_3, G_4$, and define

$$\mathbb{W}_{\mathcal{Q}}\left(G_1, G_2 \| G_3, G_4\right) = \sum_{x_i \in G_1} \sum_{x_j \in G_2} \sum_{x_k \in G_3} \sum_{x_l \in G_4} \frac{\mathbb{I}_{(i,j,k,l) \in \mathcal{Q}} - \mathbb{I}_{(k,l,i,j) \in \mathcal{Q}}}{|G_1| \, |G_2| \, |G_3| \, |G_4|}. \tag{4}$$

The idea is that clusters $G_1, G_2$ are more similar to each other than $G_3, G_4$ if their objects are, on average, more similar to each other than the objects of $G_3$ and $G_4$. This formulation suggests that an agglomerative clustering should merge $G_1, G_2$ before $G_3, G_4$ if $\mathbb{W}_{\mathcal{Q}}\left(G_1, G_2 \| G_3, G_4\right) > 0$. Also, note that $\mathbb{W}_{\mathcal{Q}}\left(G_1, G_2 \| G_3, G_4\right) = -\mathbb{W}_{\mathcal{Q}}\left(G_3, G_4 \| G_1, G_2\right)$ and $\mathbb{W}_{\mathcal{Q}}\left(G_1, G_2 \| G_1, G_2\right) = 0$, which hints that (4) is a preference relation between pairs of clusters. We use the above preference relation $\mathbb{W}_{\mathcal{Q}}$ to define a new cluster-level similarity function $W$ that can be used in Algorithm 1. Hence, given two clusters $G_p, G_q$, $p \neq q$, we define their similarity as

$$W\left(G_p, G_q\right) = \sum_{r,s=1, r \neq s}^{K} \frac{\mathbb{W}_{\mathcal{Q}}\left(G_p, G_q \| G_r, G_s\right)}{K(K-1)}. \tag{5}$$

The idea is that two clusters $G_p$ and $G_q$ are similar to each other if, on average, the pair is often preferred over the other possible cluster pairs. The above measure $W$ provides an average linkage approach based on quadruplets (4–AL), whose statistical guarantees are presented below.

**Theorem 4 (Exact recovery of planted hierarchy by 4–AL with passive comparisons).** *Let $\eta \in (0,1)$ and $\Delta = \frac{\delta}{2\sigma} e^{-L^2 \delta^2 / 4\sigma^2}$. Assume the following:*
*(i) An initial step partitions $\mathcal{X}$ into pure clusters of sizes in the range $[m, 2m]$ for some $m \leq \frac{1}{2} N_0$.*
*(ii) $\mathcal{Q}$ is a passively obtained set of quadruplet comparisons, where each tuple $(i, j, k, l)$ is observed independently with probability $p > \dfrac{C}{m\Delta^2} \max\left\{\ln N, \dfrac{1}{m} \ln\left(\dfrac{1}{\eta}\right)\right\}$ for some constant $C > 0$.*

*Then, with probability $1 - \eta$, starting from the given initial partition and using $|\mathcal{Q}| \leq pN^4$ number of passive comparisons, 4–AL exactly recovers the planted hierarchy.*

*In particular, if $L = \mathcal{O}(1)$, the above statement implies that, when $\frac{\delta}{\sigma}$ is a constant, 4–AL exactly recovers the planted hierarchy with probability $1 - \eta$ using $\mathcal{O}\left(\frac{N^4 \ln N}{m}\right)$ passive comparisons.*

Compared to 4K–AL (Theorem 3), the guarantee for 4-AL in Theorem 4 additionally requires an initial partitioning of $\mathcal{X}$ into small pure clusters of size $m$. This is reasonable in the context of the hierarchical clustering literature since existing consistency results for average linkage also require similar assumptions (Cohen-Addad et al., 2018, Theorem 5.8). In principle, one may use passive 4K–AL to obtain these initial clusters. Theorem 4 shows that if the size of initial clusters is much larger than $\ln N$, then we do not need to observe all the quadruplets. Moreover, if $L = \mathcal{O}(1)$ and we have $\Omega(N_0)$-sized initial clusters, then the subsequent steps of 4–AL require only $\mathcal{O}(N^3 \ln N)$ passive comparisons out of the $\mathcal{O}(N^4)$ total number of available comparisons. This is less quadruplets than 4K–AL, but it is still large for practical purposes. It remains an open question whether better sampling rates can be achieved in the passive case. From a practical perspective, our experiments in Section 4 demonstrate that 4–AL performs better than 4K–AL even when no initial clusters are provided, that is $m = 1$.

## 4 Experiments

In this section we evaluate our approaches on several problems: we empirically verify our theoretical findings, we compare our methods[1] to ordinal embedding based approaches on standard datasets, and we illustrate their behaviour on a comparison-based dataset.

### 4.1 Planted hierarchical model

We first use the planted hierarchical model presented in Section 2.2 to generate data and study the performance of the various methods introduced in Section 3.

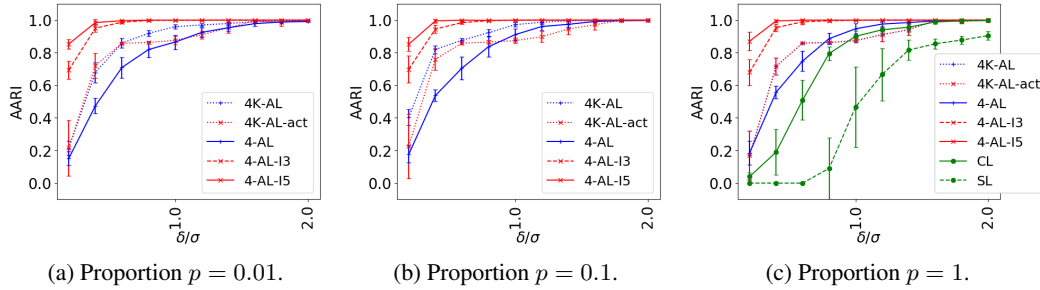

(a) Proportion $p = 0.01$.　　　(b) Proportion $p = 0.1$.　　　(c) Proportion $p = 1$.

Figure 2: AARI of the proposed methods (higher is better) on data obtained from the planted hierarchical model with $\mu = 0.8$, $\sigma = 0.1$, $L = 3$, $N_0 = 30$. In Figure 2a, 2b, and, 2c, the methods get different proportions $p$ of all the quadruplets. Best viewed in color.

**Data.** Recall that our generative model has several parameters, the within-cluster mean similarity $\mu$, the variance $\sigma^2$, the separability constant $\delta$, the depth of the planted partition $L$ and the number of examples in each cluster $N_0$. From the different guarantees presented in Section 3, it is clear that the hardness of the problem depends mainly on the signal-to-noise ratio $\frac{\delta}{\sigma}$, and the probability $p$ of observing samples for the passive methods. Hence, to study the behaviour of the different methods with respect to these two quantities, we set $\mu = 0.8$, $\sigma = 0.1$, $N_0 = 30$, and $L = 3$ and we vary $\delta \in \{0.02, 0.04, \ldots, 0.2\}$ and $p \in \{0.01, 0.02, \ldots, 0.1, 1\}$.

**Methods.** We study SL, CL, which always use the same number of active comparisons and thus are not impacted by $p$. We also consider 4K–AL with passive comparisons and its active counterpart, 4K–AL–act, implemented as described in (2) with $q = \frac{\ln N}{N}$ and the number of references in $\mathcal{R}$ chosen so that the number of comparisons observed is the same as for the passive methods. Finally, we study 4–AL with no initial pure clusters and two variants 4–AL–I3 and 4–AL–I5 that have access to initial clusters of sizes 3 and 5 respectively. These initial clusters were obtained by uniformly sampling without replacement from the $N_0$ examples contained in each of the $2^L$ ground-truth clusters.

**Evaluation function.** As a measure of performance we use the Averaged Adjusted Rand Index (AARI) between the ground truth hierarchy and the hierarchies learned by the different methods. The main idea behind the AARI is to extend the Adjusted Rand Index (Hubert and Arabie, 1985) to hierarchies by averaging over the different levels (see the appendix for a formal definition). This measure takes values in $[0, 1]$ with higher values for more similar hierarchies—AARI $(\mathcal{C}, \mathcal{C}') = 1$ implies identical hierarchies. We report the mean and the standard deviation of 10 repetitions.

**Results.** In Figure 2 we present the results for $p = 0.01$, $p = 0.1$ and $p = 1$. We defer the other results to the appendix. Firstly, similar to the theory, SL can hardly recover the planted hierarchy, even for large values of $\frac{\delta}{\sigma}$. CL performs better than SL which implies that it might be possible to derive better guarantees. We observe that 4K–AL, 4K–AL–act, and, 4–AL are able to exactly recover the true hierarchy for smaller signal-to-noise ratio and their performances do not degrade much when the number of sampled comparisons is reduced. Finally, as expected, the best method is 4–AL–I5. It uses large initial clusters but recovers the true hierarchy even for very small values of $\frac{\delta}{\sigma}$.

## 4.2 Standard clustering datasets

In this second set of experiments we compare our passive methods, 4K–AL with passive comparisons and 4–AL without initial clusters, to two baselines that use ordinal embedding as a first step.

**Baselines.** We consider t-STE (Van Der Maaten and Weinberger, 2012) and FORTE (Jain et al., 2016), followed by a standard average linkage approach using a cosine similarity as the base metric (tSTE-AL and FORTE-AL). These two methods are parametrized by the embedding dimension $d$. Since it is often difficult to automatically tune parameters in clustering (because of the lack of ground-truth) we consider several embedding dimensions and report the best results in the main paper. In the appendix, we detail the cosine similarity and report results for other embedding dimensions.

**Data.** We evaluate the different approaches on 3 different datasets commonly used in hierarchical clustering: Zoo, Glass and 20news (Heller and Ghahramani, 2005; Vikram and Dasgupta, 2016). To

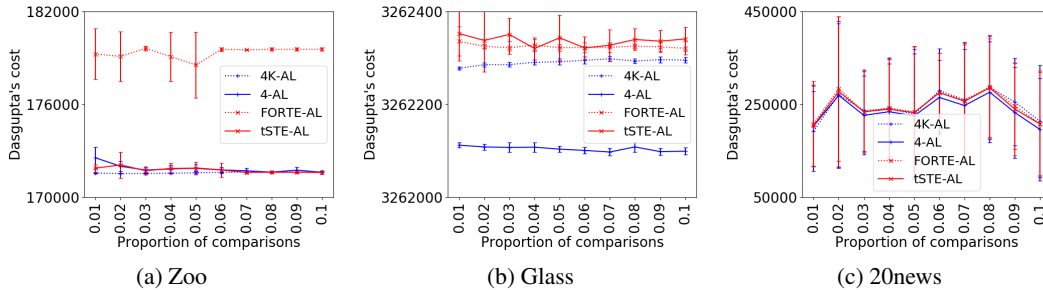

Figure 3: Dasgupta's score (lower is better) of the different methods on the Zoo, Glass and 20news datasets. The embedding dimension for FORTE–AL and tSTE–AL is set to 2. Best viewed in color.

fit the comparison-based setting we generate the comparisons using the cosine similarity. Since it is not realistic to assume that all the comparisons are available. We use the procedure described in Section 2.3 to passively obtain a proportion $p \in \{0.01, 0.02, \dots, 0.1\}$ of all the quadruplets. Some statistics on the datasets and details on the comparisons generation are presented in the appendix.

**Evaluation function.** Contrary to the planted hierarchical model, we do not have access to a ground-truth hierarchy and thus we cannot use the AARI measure. Instead, we use the recently proposed Dasgupta's cost (Dasgupta, 2016) that has been specifically designed to evaluate hierarchical clustering methods. The idea of this cost is that similar objects that are merged higher in the hierarchy should be penalized. Hence, a lower cost indicates a better hierarchy. Details are provided in the appendix. For all the experiments we report the mean and the standard deviation of 10 repetitions.

**Results.** We report the results in Figure 3. We note that the proportion of comparisons does not have a large impact as the results are, on average, stable across all regimes. Our methods are either comparable or better than the embedding-based ones. They do not need to first embed the examples and thus do not impose a strong Euclidean structure on the data. The impact of this structure is more or less pronounced depending on the dataset. Furthermore, as illustrated in the appendix, a poor choice of embedding dimension can drastically worsen the results of the embedding methods.

**Comparison-based dataset.** In the appendix, we also apply the different methods to a comparison-based dataset called Car (Kleindessner and von Luxburg, 2017b).

## 5  Conclusion

We investigated the problem of hierarchical clustering in a comparison-based setting. We showed that the single and complete linkage algorithms (SL and CL) could be used in the setting where comparisons are actively queried, but with poor exact recovery guarantees under a planted hierarchical model. We also proposed two new approaches based on average linkage (4K–AL and 4–AL) that can be used in the setting of passively obtained comparisons with good guarantees in terms of exact recovery of the planted hierarchy. An active version of 4K–AL achieves exact recovery using only $\mathcal{O}\left(N \ln N\right)$ active comparisons. Empirically, we confirmed our theoretical findings and compared our methods to two ordinal embedding based baselines on standard and comparison-based datasets.

The paper leaves several open problems. From an algorithmic perspective, the key question is whether one can develop similar provable methods in the triplet setting, where one has access to comparisons of the form "$x_i$ is more similar to $x_j$ than to $x_k$". An equivalent to passive 4K–AL can obtained using the triplet kernel of Kleindessner and von Luxburg (2017a), while triplet-based variants of active 4K–AL and 4–AL require careful designing. We leave the description of such algorithms and their theoretical analysis under planted hierarchy to a follow-up work. From a theoretical perspective, the main question is to derive necessary conditions and query complexities for exact recovery of planted hierarchy, and subsequently, validate whether the proposed algorithms are indeed optimal. Additionally, it would be interesting to analyse the performance of the proposed methods in terms of Dasgupta's score, and in presence of noisy queries, that is when some answers are randomly flipped.

**Acknowledgments**

This work has been supported by the Institutional Strategy of the University of Tübingen (Deutsche Forschungsgemeinschaft, DFG, ZUK 63), by the DFG Cluster of Excellence "Machine Learning – New Perspectives for Science", EXC 2064/1, project number 390727645, by the BMBF through the Tuebingen AI Center (FKZ: 01IS18039A), and by the Baden-Württemberg Eliteprogramm for Postdocs.

## Footnotes

[1]The code of our methods is available at `https://github.com/mperrot/ComparisonHC`.

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
