[Supplementary Material]

# Foundations of Comparison-Based Hierarchical Clustering – Appendix

**Debarghya Ghoshdastidar**[*]
Department of Informatics, TU Munich
ghoshdas@in.tum.de

**Michaël Perrot**[*]
Max Planck Institute for Intelligent Systems
michael.perrot@tuebingen.mpg.de

**Ulrike von Luxburg**
Department of Computer Science, University of Tübingen
Max Planck Institute for Intelligent Systems
luxburg@informatik.uni-tuebingen.de

Appendix A contains the proofs of our different theorems, and Appendix B has details on the experiments along with further numerical results.

## A  The hierarchical model and proofs of the theoretical results

In this section, we illustrate the planted hierarchical model and we provide detailed proofs of Theorems 1–4.

### A.1  Notations

We first recall some of the key quantities associated with the planted model, which include:

- $N$, the number of objects;
- $L$, the number of levels in the hierarchy;
- $N_0 = \frac{N}{2^L}$, the size of the pure clusters;
- $\mu$, expected similarity between pairs belonging to a pure cluster;
- $\delta$, the separation between the expected similarities across consecutive levels; and
- $\sigma$, the standard deviation of the similarities.

Throughout the appendix, we use $Z$ to denote a generic standard normal random variable, that is, $Z \sim \mathcal{N}(0, 1)$. We also define $\ell_{ij}^{lca} = \ell^{lca}(x_i, x_j)$ as the level of the ground truth tree in which the least common ancestor ($lca$) of $x_i$ and $x_j$ resides. We extend this definition to the level of $lca$ of two clusters $G, G'$, denoted by $\ell^{lca}(G, G')$. If $G, G'$ are both subsets of the same pure cluster, we write $\ell^{lca}(G, G') = L$. Hence, the range of $\ell^{lca}$ is $\{0, 1, \ldots, L\}$.

### A.2  Analysis of Single Linkage (SL) and Complete Linkage (CL)

**Proposition 1 (Active query complexity of SL and CL).** *The SL and CL algorithms require at least $\Omega(N^2)$ and at most $\mathcal{O}(N^2 \ln N)$ number of active quadruplet comparisons.*

*Proof.* In the first step of SL or CL, the algorithm merges the pair $x_i, x_j$ if $w_{ij} \geq w_{kl}$ for all $k, l \in \{1, \ldots, N\}$. This requires $\binom{N}{2}$ number of ordinal comparisons to find the minimum, and hence, the active query complexity of SL and CL is at least $\Omega(N^2)$.

---

[*]Both authors contributed equally to the paper.

To prove an upper bound on the active query complexity, it suffices to observe that single/complete linkage only requires a total ordering of the $\binom{N}{2}$ scalar similarities $\{w_{ij} : i < j\}$. Using a sorting algorithm such as merge-sort, this ordering can be easily obtained from $\mathcal{O}\left(N^2 \ln N\right)$ actively chosen comparisons. $\qquad\square$

**Theorem 1 (Exact recovery of planted hierarchy by SL and CL).** *Assume that $\eta \in (0,1)$. If $\frac{\delta}{\sigma} \geq 4\sqrt{\ln\left(\frac{N}{\eta}\right)}$, then SL and CL exactly recover the planted hierarchy with probability $1 - \eta$.*

*Conversely, for $\frac{\delta}{\sigma} \leq \frac{1}{4}\sqrt{\ln\left(\frac{N}{2^L}\right)}$ and large $\frac{N}{2^L}$, SL fails to recover the hierarchy with probability $\frac{1}{2}$.*

*Proof.* **We first prove the sufficient condition for exact recovery.** Let $Z \sim \mathcal{N}(0,1)$. It can be easily verified that $\mathbf{P}(|Z| \geq t) \leq \sqrt{\frac{2}{\pi}} \frac{1}{t} \exp(-0.5t^2)$. For $t \geq 1$, we may simply bound this by $\exp(-0.5t^2)$. Now, observe that for every $i \neq j$, $\frac{w_{ij} - \mu_{ij}}{\sigma} \sim \mathcal{N}(0,1)$. Using this, we can write

$$\mathbf{P}\left(\bigcup_{i \neq j}\left\{|w_{ij} - \mu_{ij}| \geq \frac{\delta}{2}\right\}\right) \leq \sum_{i \neq j} \mathbf{P}\left(|Z| \geq \frac{\delta}{2\sigma}\right) \leq N^2 \exp\left(-\frac{\delta^2}{8\sigma^2}\right)$$

since $\delta > 2\sigma$ under stated condition. The above probability is smaller than $\eta$ for $\delta \geq 4\sigma\sqrt{\ln(\frac{N}{\eta})}$. Thus, under the stated condition, $|w_{ij} - \mu_{ij}| < \frac{\delta}{2}$ for all $i \neq j$. We now show that the above scenario leads to exact recovery of the hierarchy by single or complete linkage clustering. Note that

$$\mathbf{E}[w_{ij}] = \mu_{ij} = \mu - (L - \ell_{ij}^{lca})\delta$$

Due to the concentration of the similarity score $w$, we know that $w_{ij}$ lies in the range $\left(\mu - (L - \ell_{ij}^{lca})\delta - \frac{\delta}{2}, \mu - (L - \ell_{ij}^{lca})\delta + \frac{\delta}{2}\right)$ for all $i \neq j$ with probability $1 - \eta$. Thus, the similarity scores corresponding to the different levels of the ground truth do not overlap, and this ensures that the agglomerative algorithms merge objects or clusters in the same order as prescribed by the ground truth. For instance, at the first stage, where the goal is to extract the pure clusters, we have $w_{ij} > \mu - \frac{\delta}{2}$ if $x_i, x_j$ belong to the same pure cluster, and $w_{ij} < \mu - \frac{\delta}{2}$ otherwise. Hence, both single and complete linkage merge objects in the same cluster first before merging objects from different clusters. The same argument also holds for the subsequent levels and hence, the claim.

**We now prove the converse statement for SL.** We first prove the result for $L = 1$. The argument easily extends to $L > 1$ from the observation that exact recovery of the entire hierarchy involves exact recovery for pairs of clusters at $L - 1$ levels. For $L = 1$, there are two pure clusters, $\mathcal{G}_1$ and $\mathcal{G}_2$, that are split at the top level of the true hierarchy.

Recall that single linkage corresponds to a cluster tree on the set of items (Chaudhuri et al., 2014). For any $t \in \mathbb{R}$, we consider the subgraph $G_t$ of the cluster tree with edge set $E_t = \{(i, j) : w_{ij} > t\}$. Observe that $G_t$ is equivalent to a stochastic block model, where

$$\mathbf{P}\left((i,j) \in E_t\right) = \begin{cases} 1 - \Phi\left(\dfrac{t - \mu}{\sigma}\right) & \text{for } i, j \text{ in the same cluster, and} \\ 1 - \Phi\left(\dfrac{t - \mu + \delta}{\sigma}\right) & \text{when } i, j \text{ belong to different clusters.} \end{cases} \tag{1}$$

Let $p, q$ denote the aforementioned within and inter-cluster edge probabilities in (1), and recall the bounds on the Gaussian tail

$$\frac{1}{\sqrt{2\pi}} \frac{1}{2x} e^{-x^2/2} < 1 - \Phi(x) < \frac{1}{\sqrt{2\pi}} \frac{1}{x} e^{-x^2/2}, \tag{2}$$

which is valid for all $x \geq 1$. Setting $t = \mu + \sigma\sqrt{2 \ln N_0}$, it is easy to verify that

$$p < \frac{1}{\sqrt{2\pi}} \frac{1}{N_0 \sqrt{2 \ln N_0}} \qquad \text{and} \qquad q > \frac{1}{\sqrt{2\pi}} \frac{1}{2\left(\sqrt{2 \ln N_0} + \frac{\delta}{\sigma}\right)} e^{-\left(\sqrt{2 \ln N_0} + \frac{\delta}{\sigma}\right)^2/2}.$$

Assuming $\frac{\delta}{\sigma} < \frac{1}{4}\sqrt{\ln N_0}$, the lower bound on $q$ can be simplified as $q > \frac{1}{10 N_0 \sqrt{N_0 \ln N_0}}$. Hence, for large enough $N_0$, we have $p < \frac{1}{N_0}$ and $q \gg \frac{\ln N_0}{N_0^2}$. Now observe that the two subgraphs of

$G_t$ restricted to $\mathcal{G}_1$ and $\mathcal{G}_2$, $G_{t|\mathcal{G}_1}$ and $G_{t|\mathcal{G}_2}$, are Erdős-Rényi graphs, each with $N_0$ vertices and edge-probability $p$. Using a standard result for random graphs (Chapter 8 of Blum et al., 2018), we can conclude that both $G_{t|\mathcal{G}_1}$ and $G_{t|\mathcal{G}_2}$ are disconnected with high probability for $p < \frac{1}{N_0}$. Similarly, since $q \gg \frac{\ln N_0}{N_0^2}$, one can conclude that, with high probability, there exist edges between $G_{t|\mathcal{G}_1}$ and $G_{t|\mathcal{G}_2}$. Based on the cluster tree perspective of single linkage (Chaudhuri et al., 2014), the above conclusions about connectivity of $G_{t|\mathcal{G}_1}$ and $G_{t|\mathcal{G}_2}$ implies that SL merges items from $\mathcal{G}_1$ and $\mathcal{G}_2$ before extracting the pure clusters. For large enough $N_0$, the probability of this event is greater than $\frac{1}{2}$. $\qquad\square$

### A.3 Analysis of Active Quadruplets Kernel based Average Linkage (4K–AL)

Recall that the active quadruplet kernel is defined in the following way. A pair of distinct items $(i_0, j_0)$ is chosen uniformly, and a set of landmark points $\mathcal{S}$ is constructed such that every $k \in \{1, \ldots, N\}$ is independently added to $\mathcal{S}$ with probability $q$. The kernel $K$ is defined as

$$K_{ij} = \sum_{k \in \mathcal{S} \setminus \{i,j\}} \left( \mathbb{I}_{\left(w_{ik} > w_{i_0 j_0}\right)} - \mathbb{I}_{\left(w_{ik} < w_{i_0 j_0}\right)} \right) \left( \mathbb{I}_{\left(w_{jk} > w_{i_0 j_0}\right)} - \mathbb{I}_{\left(w_{jk} < w_{i_0 j_0}\right)} \right) \tag{3}$$

for $i \neq j$. For ease of notation, we introduce the terms $w^* = w_{i_0 j_0}$ and $\xi_k = \mathbb{I}_{(k \in \mathcal{S})}$. It follows that $\xi_1, \ldots, \xi_N \sim_{iid}$ Bernoulli$(q)$ and, with these notations, we write the kernel function (3) as

$$K_{ij} = \sum_{k \neq i,j} \xi_k \left( 2\mathbb{I}_{(w_{ik} > w^*)} - 1 \right) \left( 2\mathbb{I}_{(w_{jk} > w^*)} - 1 \right),$$

where the re-arrangement of indicators are under the planted model assumption since any two similarity scores are distinct with probability 1 due to the Gaussian assumption.

We now restate and prove the exact recovery guarantee for 4K–AL with actively obtained comparisons.

**Theorem 2 (Exact recovery of planted hierarchy by 4K–AL with active comparisons).** *Let $\eta \in (0,1)$ and $\Delta = \frac{\eta^2}{100} \frac{\delta}{\sigma} e^{-2L^2 \delta^2 / \sigma^2}$. There exists an absolute constant $C > 0$ such that if $N_0 > \frac{4}{\Delta}\sqrt{N}$ and we set*

$$q > \max \left\{ C \frac{2^{2L}}{N\Delta^4} \ln \left( \frac{N}{\eta} \right), \frac{3}{N} \ln \left( \frac{2}{\eta} \right) \right\},$$

*then with probability at least $1 - \eta$, 4K–AL exactly recovers the planted hierarchy using at most $2qN^2$ number of actively chosen quadruplet comparisons.*

*In particular, if $L = \mathcal{O}(1)$, the above statement implies that even with $\frac{\delta}{\sigma}$ constant, 4K–AL exactly recovers the planted hierarchy with probability $1 - \eta$ using only $\mathcal{O}(N \ln N)$ active comparisons.*

*Proof.* We prove the result by proving the following statements:

- the probability that 4K–AL queries more than $2qN^2$ comparisons is at most $\frac{\eta}{2}$, and

- the probability of not achieving exact recovery is at most $\frac{\eta}{2}$.

**To derive the bound on the number of comparisons**, we observe that evaluation of the entire kernel matrix requires quadruplet comparisons of the form $\mathbb{I}_{(w_{ik} > w^*)}$ for all $i = 1, \ldots, N$ and $k \in \mathcal{S}$. Hence, the total number of comparisons is $N|\mathcal{S}|$, which can be bounded by showing that the size of $\mathcal{S}$ is at most $2qN$. This follows from Bernstein's inequality since

$$\mathbf{P}(|\mathcal{S}| > 2qN) = \mathbf{P}\left( \sum_{k=1}^{N} \xi_k - qN > qN \right)$$

$$\leq \exp\left( -\frac{q^2 N^2}{2Nq(1-q) + \frac{2}{3}qN} \right) \leq \exp\left( -\frac{qN}{3} \right),$$

which is bounded by $\frac{\eta}{2}$ since $q > \frac{3}{N} \ln \left( \frac{2}{\eta} \right)$.

**To derive the exact recovery guarantee**, we analyze the kernel matrix $K$, and also 4K–AL, conditioned on $w^*$. For this, we need to characterize the behaviour of $w^*$ under the planted model. Since $w^*$ is the similarity of a randomly chosen pair, one can observe that $w^* \sim \sum_{\ell=0}^{L} a_\ell \mathcal{N}\left(\mu - \ell\delta, \sigma^2\right)$ has a mixture of Gaussian distribution, where the weights $a_0 = \frac{2^L \binom{N_0}{2}}{\binom{N}{2}}$ and $a_\ell = \frac{2^{L+\ell-2} N_0^2}{\binom{N}{2}}$ for $\ell = 1, \ldots, L$ are the proportion of similarities corresponding to item pairs merged at level $(L - \ell)$ of the panted hierarchy. We claim that, with probability $1 - \frac{\eta}{4}$,

$$\mu - L\delta - \sigma\sqrt{2\ln\left(\frac{8}{\eta}\right)} < w^* < \mu + \sigma\sqrt{2\ln\left(\frac{8}{\eta}\right)}. \tag{4}$$

The bounds follow from the mixture of Gaussian nature of $w^*$ since

$$\mathbf{P}\left(w^* > t\right) = \sum_{\ell=0}^{L} a_\ell \mathbf{P}\left(\mu - \ell\delta + \sigma Z > t\right)$$
$$\leq \mathbf{P}(\mu + \sigma Z > t),$$

where we use $Z$ to denote a standard normal random variable. Setting $t = \mu + \sigma\sqrt{2\ln\left(\frac{8}{\eta}\right)}$ and using the upper bound on Gaussian tail probability (2), we can bound the above probability by $\frac{\eta}{8}$. A similar argument holds for the lower bound on $w^*$, where the probability of violating the bound is also at most $\frac{\eta}{8}$. Hence, the bounds in (4) hold with probability $1 - \frac{\eta}{4}$.

We next compute the expected kernel matrix (3) conditioned on the knowledge of $w^*$. For this, we first define the quantities

$$\beta_{\ell,w^*} = 2\mathbf{P}_{Z\sim\mathcal{N}(0,1)}\left(\mu - (L-\ell)\delta + \sigma Z > w^* \middle| w^*\right) - 1, \text{ and}$$
$$\beta_\ell = 2\mathbf{P}_{Z,Z'\sim\mathcal{N}(0,1)}\left(\mu + \sigma Z > \mu - \ell\delta + \sigma Z'\right) - 1 = 2\Phi\left(\frac{\ell\delta}{\sqrt{2}\sigma}\right) - 1 \tag{5}$$

for any $\ell \in \mathbb{R}$ and $w^* \in \mathbb{R}$. Observe that $\beta_{\ell,w^*} = \mathbf{E}\left[2\mathbb{I}_{(w_{ij}>w^*)} - 1 \middle| w^*\right]$ when $\ell_{ij}^{lca} = \ell$, whereas $\beta_\ell = \mathbf{E}\left[2\mathbb{I}_{(w_{ij}>w_{kl})} - 1\right]$ when $\ell_{ij}^{lca} - \ell_{kl}^{lca} = \ell$. In particular, $\beta_0 = 0$. Based on (5) and the observation that the product terms in (3) are independent conditioned on $w^*$, we write for any $i \neq j$,

$$\mathbf{E}\left[K_{ij}\middle|w^*\right] = \sum_{k\neq i,j} q\beta_{\ell_{ik}^{lca},w^*}\beta_{\ell_{jk}^{lca},w^*}.$$

Recall that, under the planted hierarchy, $\mathcal{X}$ is partitioned in pure clusters $\mathcal{G}_1, \ldots, \mathcal{G}_{2^L}$. We abuse notation to write $\mathcal{G}_r$ as the set $\{i : x_i \in \mathcal{G}_r\}$. In (3), observe that each term in the sum depends only on the groups containing $i, j, k$, and hence, we may only compute it for each group and multiply by the number of terms in the group. If $i, j \in \mathcal{G}_1$, then $k$ can take only $(N_0 - 2)$ values in $\mathcal{G}_1$, and $N_0$ values in other groups. We may perform the entire computation only at group level, and then use a multiplicative factor of $(1 \pm \epsilon)$ with $\epsilon = \frac{4}{N_0}$ to account for fluctuations in the number of terms from each group. Here, $\mathbf{E}[K_{ij}|w^*] = (1 \pm \epsilon)a$ denotes $(1 - \epsilon)a \leq \mathbf{E}[K_{ij}|w^*] \leq (1 + \epsilon)a$. Allowing a fluctuation of $(1 \pm \epsilon)$ also helps to ignore the small effect of the case where $(i, k)$ or $(j, k)$ corresponds to $(i_0, j_0)$, that is, the reference pair for which $w^* = w_{i_0 j_0}$. Thus, for $i, j$ such that $i \neq j$ and $\ell_{ij}^{lca} = \ell$, we have

$$\mathbf{E}[K_{ij}|w^*] = (1 \pm \epsilon)qN_0 \sum_{r=1}^{2^L} \beta_{\ell^{lca}(i,\mathcal{G}_r),w^*}\beta_{\ell^{lca}(j,\mathcal{G}_r),w^*}$$

$$= (1 \pm \epsilon)qN_0 \sum_{t,t'=0}^{L} \beta_{t,w^*}\beta_{t',w^*}\#\{r : \ell^{lca}(i,\mathcal{G}_r) = t, \ell^{lca}(j,\mathcal{G}_r) = t'\}, \tag{6}$$

where the second equality explicitly mentions that we need to count the number of different pure clusters that are merged with $i$ or $j$ at different levels of the true hierarchy. We now consider different

cases. First, if $i, j$ belong to same group, then $\ell = L$ and $\ell^{lca}(i, \mathcal{G}_r) = \ell^{lca}(j, \mathcal{G}_r)$ for every $r$. So,

$$\kappa_L := \mathbf{E}[K_{ij}|w^*] = (1 \pm \epsilon)qN_0 \sum_{t=0}^{L}(2^{L-1-t} \vee 1)\beta_{t,w^*}^2,$$

which we denote by a quantity $\kappa_L$ noting that it only depends on the level $L$ and not on $i, j$. Here, $\vee$ denotes the maximum of two values. The numbers of clusters are computed based on the fact that there is only one cluster at levels $L$ or $L-1$, and otherwise $2^{L-1-t}$ groups are merged with $i$ at level-$t$. If $i, j$ are not in the same group, that is, $\ell = \ell_{ij}^{lca} < L$, then we observe:

- if $t < \ell$, then for any $\mathcal{G}_r$ such that $\ell^{lca}(i, \mathcal{G}_r) = t$, we also have $\ell^{lca}(j, \mathcal{G}_r) = t$. So we may only consider cases $t = t'$ when $t < \ell$.

- there is no $\mathcal{G}_r$ such that $\ell^{lca}(i, \mathcal{G}_r) = \ell^{lca}(j, \mathcal{G}_r) = \ell$ which happens because the hierarchy is a binary tree and $\mathcal{G}_r$ must either merge first with $i$ or with $j$. So, we do not need to consider $t = t' = \ell$, which is the main difference from the case $\ell = L$.

- if $t > \ell$, then for any $\mathcal{G}_r$ with $\ell^{lca}(i, \mathcal{G}_r) = t$, we have $\ell^{lca}(j, \mathcal{G}_r) = \ell$. So we may set $t' = \ell$ whenever we have $t > \ell$. Similarly, we should also count the cases $t' > \ell, t = \ell$.

Thus, we can decompose the summation into three parts based on the conditions on $t, t'$ ($t = t' < \ell$; $t > \ell, t' = \ell$; $t = \ell, t' > \ell$). For each case, we should count $\#\{r : \ell^{lca}(i, \mathcal{G}_r) = t, \ell^{lca}(j, \mathcal{G}_r) = t'\}$. To compute these, we note that $\#\{r : \ell^{lca}(i, \mathcal{G}_r) = \ell^{lca}(j, \mathcal{G}_r) = t\} = 2^{L-1-t}$ when $t = t' < \ell$ as used above. But when $t > \ell, t' = \ell$, we have $\#\{r : \ell^{lca}(i, \mathcal{G}_r) = t, \ell^{lca}(j, \mathcal{G}_r) = \ell\} = 2^{L-1-t} \vee 1$ since this counts only those groups which merge with $i$ at level-$t$, and $t'$ plays no role in the count. A similar argument holds for the case $t = \ell, t' > \ell$. Based on this, we compute $\mathbf{E}[K_{ij}|w^*]$ for the case $\ell_{ij}^{lca} = \ell < L$ and denote the expected value by $\kappa_\ell$, noting that it does not depend on $i, j$. We have

$$\kappa_\ell := \mathbf{E}[K_{ij}|w^*] = (1 \pm \epsilon)qN_0 \left[\sum_{t=0}^{\ell-1} 2^{L-1-t}\beta_{t,w^*}^2 + 2\sum_{t=\ell+1}^{L}(2^{L-1-t} \vee 1)\beta_{t,w^*}\beta_{\ell,w^*}\right],$$

where the second term, counted twice, corresponds to both the cases of $t > \ell$ or $t' > \ell$, which behave similarly. Since $\beta_{t,w^*} \in [-1, 1]$, one can easily verify that $|\kappa_\ell| \leq qN$ for all $\ell$.

The above discussion leads to the conclusion that $\mathbf{E}[K|w^*]$ has a block diagonal structure with exactly $L+1$ distinct off-diagonal entries, $\kappa_0, \ldots, \kappa_L$, and the block structure corresponds to the planted hierarchy shown in Figure 1 in the main paper (right). We now show that these distinct terms are sufficiently separated, that is, $\kappa_{\ell+1} - \kappa_\ell$ is large for every $\ell = 0, 1, \ldots, L-1$. To derive this, we require a lower bound on

$$\beta_{t+1,w^*} - \beta_{t,w^*} = 2\mathbf{P}\left(\mu - (L-t-1)\delta + \sigma Z > w^* \big| w^*\right) - 2\mathbf{P}\left(\mu - (L-t)\delta + \sigma Z > w^* \big| w^*\right)$$

$$= \sqrt{\frac{2}{\pi}} \int_{(w^*-\mu+(L-t-1)\delta)/\sigma}^{(w^*-\mu+(L-t)\delta)/\sigma} e^{-z^2/2}\mathrm{d}z$$

$$\geq \sqrt{\frac{2}{\pi}}\frac{\delta}{\sigma}e^{-\left(a^2\vee(a-\delta)^2\right)/2\sigma^2},$$

where $a = w^* - \mu + (L-t)\delta$. Conditioned on the bounds $w^*$ stated in (4), one can see that

$$a^2 \vee (a-\delta)^2 < 2(L+1)^2\delta^2 + 4\sigma^2 \ln\left(\frac{8}{\eta}\right),$$

where we use the fact that $t \in [0, L]$ and the inequality $(x+y)^2 \leq 2(x^2+y^2)$. Plugging this into the above derivation shows that $\beta_{t+1,w^*} - \beta_{t,w^*} > \Delta$ for any $t \in [0, L]$, where $\Delta$ is defined in the statement of theorem. We use the above bound to show that

$$\kappa_L - \kappa_{L-1} > qN_0 \left(\beta_{L,w^*}^2 - \beta_{L-1,w^*}^2\right)^2 - 2\epsilon qN$$
$$> qN_0\Delta^2 - q2^{L+3},$$

where the second term, involving $\epsilon$, takes care of the fluctuation due to our approximate computations of $\kappa_\ell$ and is simply bounded by the upper bound on $\kappa_\ell$. Similarly, for any $\ell < L - 1$,

$$\kappa_{\ell+1} - \kappa_\ell > qN_0 \Bigg[ 2^{L-1-\ell}\beta_{\ell,w^*}^2 - 2^{L-1-\ell}\beta_{\ell,w^*}\beta_{\ell+1,w^*}$$

$$+ 2\sum_{t=\ell+2}^{L}(2^{L-1-t}\vee 1)\beta_{t,w^*}(\beta_{\ell+1,w^*} - \beta_{\ell,w^*}) \Bigg] - 2\epsilon qN$$

$$= qN_0 2\sum_{t=\ell+2}^{L}(2^{L-1-t}\vee 1)(\beta_{t,w^*} - \beta_{\ell,w^*})(\beta_{\ell+1,w^*} - \beta_{\ell,w^*}) - 2\epsilon qN$$

$$> 2^{L-\ell-1}qN_0\Delta^2 - q2^{L+3},$$

where the equality follows since $2^{L-1-\ell} = 2\sum_{t=\ell+2}^{L}(2^{L-1-t}\vee 1)$, and subsequently, we note that $\beta_{t,w^*} - \beta_{\ell,w^*} > \beta_{\ell+1,w^*} - \beta_{\ell,w^*} > \Delta$ for all $t \geq \ell+2$. Hence, we can conclude that for $N_0 > \frac{4}{\Delta}\sqrt{N}$, or equivalently, $N_0 > \frac{2^{L+4}}{\Delta^2}$,

$$\kappa_{\ell+1} - \kappa_\ell > \frac{qN_0\Delta^2}{2} \tag{7}$$

for all $\ell = 0, 1, \ldots, L - 1$. We subsequently show that under the condition on $q$ assumed in the theorem, with probability $1 - \frac{\eta}{4}$,

$$K_{ij} - \mathbf{E}[K_{ij}|w^*] < \frac{qN_0\Delta^2}{4} \tag{8}$$

for all $i \neq j$. This implies that all random entries of $K$ corresponding to different levels of hierarchy in the ground truth tree are non-overlapping. Hence, one can simply use the arguments in the proof of Theorem 1 to show that average linkage (or even single/complete linkage) recovers the planted hierarchy. We complete the proof by deriving the concentration result of (8). From (3), we observe that, conditioned on $w^*$, the entry $K_{ij}$ is a sum of $N - 2$ independent random variables each lying in the range $[-1, 1]$. Hence, a direct application of Bernstein's inequality implies that

$$\mathbf{P}\left(|K_{ij} - \mathbf{E}[K_{ij}|w^*]| > \sqrt{3qN\ln\left(\frac{4N^2}{\eta}\right)} \bigvee 3\ln\left(\frac{4N^2}{\eta}\right) \Bigg| w^*\right) \leq \frac{\eta}{2N^2}.$$

Using the symmetry of $K$ and the union bound, it follows that the above entry-wise concentration holds for all $i \neq j$ with probability at least $1 - \frac{\eta}{4}$. Finally, for $q > C\frac{2^{2L}}{N\Delta^4}\ln\left(\frac{N}{\eta}\right)$ with $C > 0$ large enough, it is easy to verify that $\frac{1}{4}qN_0\Delta^2$ is larger than the deviation obtained using Bernstein's inequality. The above argument leads to the claim of Theorem 2.

**To verify the claim for fixed $L$ and $\frac{\delta}{\sigma}$,** we note that, in this case, $\Delta$ is constant and $N_0 = \Omega(N)$. Hence, using $q = \frac{c\ln N}{N}$ for a large enough constant $c$ immediately leads to the exact recovery guarantee and number of comparisons. $\square$

### A.4 Analysis of Passive Quadruplets Kernel based Average Linkage (4K–AL)

In the passive setting, we do not have the freedom of querying specific comparisons but have access to only a pre-computed set of quadruplet comparisons $\mathcal{Q} \subset \{(i, j, k, l) : w_{ij} > w_{kl}\}$. Hence, we use a variant of the kernel in (3), which relies only on passively obtained comparisons.

$$K_{ij} = \sum_{\substack{k,l=1 \\ k<l}}^{N}\sum_{r=1}^{N}\left(\mathbb{I}_{(i,r,k,l)\in\mathcal{Q}} - \mathbb{I}_{(k,l,i,r)\in\mathcal{Q}}\right)\left(\mathbb{I}_{(j,r,k,l)\in\mathcal{Q}} - \mathbb{I}_{(k,l,j,r)\in\mathcal{Q}}\right). \tag{9}$$

In principle, the above kernel extends the actively computed kernel (3) by using all $\binom{N}{2}$ pairs of $(k, l)$ as references in comparison to only one used in (3). However, each term in the sum only contributes

when we simultaneously observe the comparisons between $(i, r)$ and $(k, l)$ and between $(j, r)$ and $(k, l)$.

In the following, we assume that the model for obtaining passive comparisons is the one described in Section 2.3 of the main paper. For every tuple $(i, r, k, l)$, we assume that with probability $p \in (0, 1]$, there is a comparison $w_{ir} \gtrless w_{kl}$ and based on the comparison either $(i, r, k, l) \in \mathcal{Q}$ or $(k, l, i, r) \in \mathcal{Q}$. We also assume that the observation of the quadruplet comparisons are independent. Based on this model, we define a set of i.i.d. Bernoullis $\{\xi_{irkl} \sim \text{Bernoulli}(p) : i, r, k, l \text{ such that } i < r, k < l, (i, r) < (k, l)\}$, where we order the indices/ index pairs to avoid repeated counting of the same tuple. It follows that $|\mathcal{Q}| = \sum_{i,r,k,l} \xi_{irkl}$, and from Bernstein's inequality, it follows that $|\mathcal{Q}| = \mathcal{O}\left(pN^4\right)$ with high probability. Using this notation, we may re-write the kernel function in (9) as

$$K_{ij} = \sum_{k<l} \sum_{r \neq i,j} \xi_{irkl}\xi_{jrkl} \left(\mathbb{I}_{(w_{ir}>w_{kl})} - \mathbb{I}_{(w_{ir}<w_{kl})}\right) \left(\mathbb{I}_{(w_{jr}>w_{kl})} - \mathbb{I}_{(w_{jr}<w_{kl})}\right).$$

We now restate and prove the exact recovery guarantee for average linkage with the aforementioned kernel.

**Theorem 3 (Exact recovery of planted hierarchy by 4K–AL with passive comparisons).** *Let $\eta \in (0, 1)$ and $\Delta = \frac{\delta}{2\sigma}e^{-L^2\delta^2/4\sigma^2}$. There exists an absolute constant $C > 0$ such that if $N_0 > \frac{8}{\Delta}\sqrt{N}$ and we set*

$$p > \max\left\{C\frac{2^L}{\Delta^2}\sqrt{\frac{1}{N}\ln\left(\frac{N}{\eta}\right)}, \frac{2}{N^4}\ln\left(\frac{2}{\eta}\right)\right\},$$

*then with probability at least $1 - \eta$, the 4K–AL algorithm exactly recovers the planted hierarchy using at most $pN^4$ quadruplet comparisons, which are passively obtained based on the model described in Section 2.3 (of the main paper).*

*In particular, if $L = \mathcal{O}(1)$, the above statement implies that even with $\frac{\delta}{\sigma}$ constant, 4K–AL exactly recovers the planted hierarchy with probability $1 - \eta$ using $\mathcal{O}\left(N^{7/2}\ln N\right)$ passive comparisons.*

*Proof.* **The upper bound on the number of comparisons** follow by noting that $|\mathcal{Q}|$ is a sum of $\binom{\binom{N}{2}}{2}$ i.i.d. Bernoullis, and hence, the bound of $pN^4$ holds with probability $1 - \frac{\eta}{2}$ for $p > \frac{2}{N^4}\ln(\frac{2}{\eta})$.

**The proof for exact recovery** has a similar structure as that of Theorem 2, the only difference being that the analysis does not depend on a fixed reference pair. In particular, we can write the expected entries of the kernel matrix in (9) as

$$\mathbf{E}[K_{ij}] = \sum_{k<l} \sum_{r \neq i,j} p^2\left(2\mathbf{P}(w_{ir} > w_{kl}) - 1\right)\left(2\mathbf{P}(w_{jr} > w_{kl}) - 1\right)$$

$$= \frac{1}{2} \sum_{k \neq l} \sum_{r \neq i,j} p^2 \beta_{\ell_{ir}^{lca} - \ell_{kl}^{lca}} \beta_{\ell_{jr}^{lca} - \ell_{kl}^{lca}},$$

where $\beta$ is defined in (5). As in the proof of Theorem 2, we show that $\mathbf{E}[K_{ij}]$ can take at most $L + 1$ distinct values depending on the level $\ell_{ij}^{lca}$. As before, we decompose the above summation depending on $\ell_{ir}^{lca}, \ell_{jr}^{lca}$ and $\ell_{kl}^{lca}$, and also allow a fluctuation of $(1 \pm \epsilon)$ with $\epsilon = \frac{8}{N_0}$ to take care of minor effects of ignoring cases such as $k = l$ or $r = i, j$. We write the expectation in terms of the clusters as

$$\mathbf{E}[K_{ij}] = \frac{(1 \pm \epsilon)}{2}p^2 N_0^3 \sum_{r,k,l=1}^{2^L} \beta_{\ell^{lca}(i,\mathcal{G}_r)-\ell^{lca}(\mathcal{G}_k,\mathcal{G}_l)} \beta_{\ell^{lca}(j,\mathcal{G}_r)-\ell^{lca}(\mathcal{G}_k,\mathcal{G}_l)}$$

$$= \frac{(1 \pm \epsilon)}{2}p^2 N_0^3 \sum_{s,t,t'=0}^{L} \beta_{t-s}\beta_{t'-s} \times$$
$$\#\{r : \ell^{lca}(i, \mathcal{G}_r) = t, \ell^{lca}(j, \mathcal{G}_r) = t'\}\#\{k, l : \ell^{lca}(\mathcal{G}_k, \mathcal{G}_l) = s\}$$

$$= (1 \pm \epsilon)p^2 N_0^3 2^{L-1} \sum_{s,t,t'=0}^{L} (2^{L-1-s} \vee 1)\beta_{t-s}\beta_{t'-s}\#\{r : \ell^{lca}(i, \mathcal{G}_r) = t, \ell^{lca}(j, \mathcal{G}_r) = t'\}$$

The last step holds since every cluster $\mathcal{G}_l$ is merged with $(2^{L-1-s} \vee 1)$ clusters at level-$s$, and hence, $\#\{k, l : \ell^{lca}(\mathcal{G}_k, \mathcal{G}_l) = s\} = 2^L(2^{L-1-s} \vee 1)$.

We now compute $\kappa_\ell = \mathbf{E}[K_{ij}]$ where $\ell = \ell_{ij}^{lca}$. For, $\ell = L$, that is, when $i, j$ belong to the same cluster, $\ell^{lca}(i, \mathcal{G}_r) = \ell^{lca}(j, \mathcal{G}_r)$ for every cluster. Hence,

$$\kappa_L = (1 \pm \epsilon)p^2 N_0^3 2^{L-1} \sum_{s,t=0}^{L} (2^{L-1-s} \vee 1)(2^{L-1-t} \vee 1)\beta_{t-s}^2 \, .$$

For $\ell_{ij}^{lca} = \ell < L$, we have three possible cases as mentioned in the proof of Theorem 2: $(t = t' < \ell)$; $(t > \ell, t' = \ell)$; and $(t = \ell, t' > \ell)$. Decomposing the summation based on these cases and noting that $(t > \ell, t' = \ell)$ and $(t = \ell, t' > \ell)$ lead to similar terms, we have

$$\kappa_\ell = (1 \pm \epsilon)p^2 N_0^3 2^{L-1} \sum_{s=0}^{L}(2^{L-1-s} \vee 1) \left[ \sum_{t=0}^{\ell-1} 2^{L-1-t}\beta_{t-s}^2 + 2 \sum_{t=\ell+1}^{L}(2^{L-1-t} \vee 1)\beta_{t-s}\beta_{\ell-s} \right]$$

for every $\ell = 0, 1, \ldots, L-1$. We now derive a lower bound on the separation $\kappa_{\ell+1} - \kappa_\ell$, which depends on the observation that $|\kappa_\ell| \leq \frac{1}{2}p^2 N^3$ for every $\ell$, and a lower bound on

$$\beta_{t+1-s} - \beta_{t-s} \geq \min_{r \in [-L, L-1]} \beta_{r+1} - \beta_r$$

$$= \min_{r \in [-L, L-1]} \sqrt{\frac{2}{\pi}} \int_{r\delta/\sqrt{2}\sigma}^{(r+1)\delta/\sqrt{2}\sigma} e^{-z^2/2} \mathrm{d}z$$

$$> \frac{1}{\sqrt{\pi}} \frac{\delta}{\sigma} e^{-L^2\delta^2/4\sigma^2} \, .$$

The lower bound is larger than $\Delta$ stated in the theorem. Based on this bound and noting that $2^L = \sum_{s=0}^{L}(2^{L-1-s} \vee 1)$, we obtain

$$\kappa_L - \kappa_{L-1} > p^2 N_0^3 2^{L-1} \sum_{s=0}^{L}(2^{L-s-1} \vee 1)(\beta_{L-s} - \beta_{L-1-s})^2 - \epsilon p^2 N^3$$

$$> \frac{1}{2^{L+1}}p^2 N^3 \Delta^2 - p^2 N^2 2^{L+3},$$

which is at least $\frac{1}{2^{L+2}}p^2 N^3 \Delta^2$ if $N > \frac{2^{2L+5}}{\Delta^2}$, or equivalently, $N_0 > \frac{4\sqrt{2}}{\Delta}\sqrt{N}$. Similarly, for $\ell < L-1$, we have

$$\kappa_{\ell+1} - \kappa_\ell > p^2 N_0^3 2^{L-1} \sum_{s=0}^{L}(2^{L-1-s} \vee 1) \left[ 2^{L-1-\ell}\beta_{\ell-s}^2 - 2^{L-1-\ell}\beta_{\ell+1-s}\beta_{\ell-s} \right.$$

$$\left. + 2 \sum_{t=\ell+2}^{L}(2^{L-1-t} \vee 1)\beta_{t-s}(\beta_{\ell+1-s} - \beta_{\ell-s}) \right] - \epsilon p^2 N^3$$

$$> p^2 N_0^3 2^L \sum_{s=0}^{L} \sum_{t=\ell+2}^{L}(2^{L-1-s} \vee 1)(2^{L-1-t} \vee 1)\Delta^2 - p^2 N^2 2^{L+3}$$

$$= p^2 N_0^3 2^{3L-\ell-2}\Delta^2 - p^2 N^2 2^{L+3}$$

$$> \frac{p^2 N^3 \Delta^2}{2^{\ell+2}} \, .$$

The second step follows by using $2 \sum_{t=\ell+2}^{L}(2^{L-1-t} \vee 1) = 2^{L-1-\ell}$ and $\beta_{\ell+1-s} - \beta_{\ell-s} > \Delta$. The third step computes the summation, and the fourth holds when $N_0 > \frac{8}{\Delta}\sqrt{N}$. Thus for every $\ell$, we obtain a minimum separation

$$\kappa_{\ell+1} - \kappa_\ell > \frac{1}{2^{L+2}}p^2 N^3 \Delta^2.$$

Following the proof idea of Theorem 2, it only remains to show that the fluctuation of $|K_{ij} - \mathbf{E}[K_{ij}]|$ is less than half of this minimum separation for all $i < j$ since, under this scenario, one can argue that entries of $K$ corresponding to different levels of the planted hierarchy are well-separated, and hence, the planted hierarchy is exactly recovered by average linkage. Thus, to complete the proof, we derive the following concentration inequality

$$\mathbf{P}\left(|K_{ij} - \mathbf{E}[K_{ij}]| > \sqrt{2p^2 N^5 \ln\left(\frac{2N^4}{\eta}\right)} \bigvee 2N^2 \ln\left(\frac{2N^4}{\eta}\right)\right) \leq \frac{\eta}{N^2}. \qquad (10)$$

By union bound, it follows that with probability $1 - \frac{\eta}{2}$, the above bound holds for all $i < j$, whereas setting $p > C\frac{2^L}{\Delta^2}\sqrt{\frac{1}{N}\ln\left(\frac{N}{\eta}\right)}$ for $C > 0$ large enough ensures that the deviation is smaller than $\frac{1}{2^{L+3}}p^2 N^3 \Delta^2$. To derive (10), we note that $K_{ij} - \mathbf{E}[K_{ij}] = \sum_{k<l}\sum_{r\neq i,j} B_{rkl}$ is a sum of $\binom{N}{2}(N-2)$ random variables, where we use $B_{rkl} \in [-1,1]$ to denote each term in the summation. One can verify that $B_{rkl}$ has zero mean and its variance is smaller that $p^2$. Moreover, each $B_{rkl}$ is dependant on all $(N-3)$ random variables, $\{B_{r'kl} : r' \neq r\}$, and all $\binom{N}{2} - 1$ random variables, $\{B_{rk'l'} : (k',l') \neq (k,l)\}$. Hence, if we draw a dependency graph among these random variable, we obtain a regular graph with the vertex degree of each node being $(N + \binom{N}{2} - 4) < N^2$. We use the concentration technique described in Section 2.3.2 of Janson and Ruciński (2002), where the key observation is that for any graph with maximum degree $d$, one can find an equitable colouring with $d + 1$ colours, that is a colouring where all colour classes (independent sets) differ in size by at most one. In the present context, it implies that one can split the set of random variables into at most $N^2$ subsets, $\mathcal{C}_1, \ldots, \mathcal{C}_{N^2}$ such that each subset contains at most $\frac{\binom{N}{2}(N-3)}{N^2} < \frac{N}{2}$ variables that are mutually independent. Hence, we can apply union bound followed by Bernstein's inequality to write

$$\mathbf{P}\left(|K_{ij} - \mathbf{E}[K_{ij}]| > \tau\right) \leq \mathbf{P}\left(\bigcup_{s=1}^{N^2}\left|\sum_{(r,k,l)\in\mathcal{C}_s} B_{rkl}\right| > \frac{\tau}{N^2}\right)$$

$$\leq \sum_{s=1}^{N^2}\mathbf{P}\left(\left|\sum_{(r,k,l)\in\mathcal{C}_s} B_{rkl}\right| > \frac{\tau}{N^2}\right)$$

$$\leq 2N^2 \exp\left(-\frac{\frac{\tau^2}{N^4}}{p^2 N + \frac{2}{3}\frac{\tau}{N^2}}\right) \leq 2N^2 \exp\left(-\frac{\tau^2}{2p^2 N^5} \bigvee \frac{\tau}{2N^2}\right).$$

For $\tau = \sqrt{2p^2 N^5 \ln\left(\frac{2N^4}{\eta}\right)} \bigvee 2N^2 \ln\left(\frac{2N^4}{\eta}\right)$, the probability is smaller than $\frac{\eta}{N^2}$, which results in the conclusion of (10).

**To verify the claim for fixed** $L$ **and** $\frac{\delta}{\sigma}$, we note that in this case, $\Delta$ is constant and $N_0 = \Omega(N)$. Hence, using $p = c\sqrt{\frac{\ln N}{N}}$ for a large enough constant $c$ immediately leads to the exact recovery guarantee and the number of passive comparisons. $\qquad\square$

## A.5  Analysis of Quadruplets based Average Linkage (4–AL)

The proposed 4–AL algorithms estimates the relative similarity between two pairs of clusters. For instance, let $G_1, G_2, G_3, G_4$ be four clusters such that $G_1, G_2$ are disjoint and so are $G_3, G_4$, we define

$$\mathbb{W}_{\mathcal{Q}}(G_1, G_2\|G_3, G_4) = \sum_{x_i\in G_1}\sum_{x_j\in G_2}\sum_{x_k\in G_3}\sum_{x_l\in G_4}\frac{\mathbb{I}_{(i,j,k,l)\in\mathcal{Q}} - \mathbb{I}_{(k,l,i,j)\in\mathcal{Q}}}{|G_1||G_2||G_3||G_4|}. \qquad (11)$$

Based on our model for passive comparisons, where $\xi_{ijkl} \sim \text{Bernoulli}(p)$ is the indicator for observing tuple $(i, j, k, l)$, we may re-write the preference relation in (11) as

$$\mathbb{W}_{\mathcal{Q}}(G_1, G_2\|G_3, G_4) = \sum_{x_i\in G_1}\sum_{x_j\in G_2}\sum_{x_k\in G_3}\sum_{x_l\in G_4}\frac{\xi_{ijkl}(\mathbb{I}_{(w_{ij}>w_{kl})} - \mathbb{I}_{(w_{ij}<w_{kl})})}{|G_1||G_2||G_3||G_4|}.$$

Subsequently, we use the above preference relation $\mathbb{W}_{\mathcal{Q}}$ to define a similarity function $W$ in the following way. Suppose that we have a disjoint partition $G_1, \ldots, G_K$ of $\mathcal{X}$ and that we want to know which clusters should be merged next. We define the similarity of clusters $G_p, G_q$, $p \neq q$, as

$$W(G_p, G_q) = \sum_{r,s=1, r \neq s}^{K} \frac{\mathbb{W}_{\mathcal{Q}}(G_p, G_q \| G_r, G_s)}{K(K-1)}. \tag{12}$$

The underlying idea is that two clusters $G_p$ and $G_q$ are similar to each other if, on average, the pair is often preferred over the other possible cluster pairs. The above similarity measure $W$, in conjunction with the hierarchical clustering principle (Algorithm 1 in the main paper), results in the proposed 4–AL algorithm. Below, we restate and prove the exact recovery guarantee for 4–AL using passively obtained quadruplet comparisons.

**Theorem 4 (Exact recovery of planted hierarchy by 4–AL with passive comparisons).** *Let $\eta \in (0,1)$ and $\Delta = \frac{\delta}{2\sigma} e^{-L^2 \delta^2 / 4\sigma^2}$. Assume the following:*
*(i) An initial step partitions $\mathcal{X}$ into pure clusters of sizes in the range $[m, 2m]$ for some $m \leq \frac{1}{2} N_0$.*
*(ii) $\mathcal{Q}$ is a passively obtained set of quadruplet comparisons, where each tuple $(i, j, k, l)$ is observed*
*independently with probability $p > \dfrac{C}{m\Delta^2} \max\left\{ \ln N, \dfrac{1}{m} \ln\left(\dfrac{1}{\eta}\right) \right\}$ for some constant $C > 0$.*

*Then, with probability $1 - \eta$, starting from the given initial partition and using $|\mathcal{Q}| \leq pN^4$ number of passive comparisons, 4–AL exactly recovers the planted hierarchy.*

*In particular, if $L = \mathcal{O}(1)$, the above statement implies that, when $\frac{\delta}{\sigma}$ is a constant, 4–AL exactly recovers the planted hierarchy with probability $1 - \eta$ using $\mathcal{O}\left(\frac{N^4 \ln N}{m}\right)$ passive comparisons.*

*Proof.* The bound $|\mathcal{Q}| < pN^4$ with probability $1 - \frac{\eta}{2}$ is derived similarly to the bound on $|\mathcal{Q}|$ in Theorem 3. Hence, we only prove the exact recovery guarantee.

We first analyze the algorithm under expectation. Assume that at some stage of the agglomerative iterations, we have a partition $G_1, \ldots, G_K$ of $\mathcal{X}$. Assume that the partition adheres to the ground truth, that is, either each $G_p$ is a subset of a pure cluster or an union of several pure clusters that corresponds to one of the nodes in the top $L$ levels of the true hierarchy. Consider $p, q, r, s \in \{1, \ldots, K\}$ such that $p \neq q$, $r \neq s$, $\ell^{lca}(G_p, G_q) = \ell$ and $\ell^{lca}(G_r, G_s) = \ell'$. From the definition of $\mathbb{W}_{\mathcal{Q}}$, we have

$$\mathbf{E}[\mathbb{W}_{\mathcal{Q}}(G_p, G_q \| G_r, G_s)] = \sum_{x_i \in G_p} \sum_{x_j \in G_q} \sum_{x_k \in G_r} \sum_{x_l \in G_s} \frac{p\big(2\mathbf{P}(w_{ij} > w_{kl}) - 1\big)}{|G_p| |G_q| |G_r| |G_s|}$$
$$= \sum_{x_i \in G_p} \sum_{x_j \in G_q} \sum_{x_k \in G_r} \sum_{x_l \in G_s} \frac{p\beta_{\ell - \ell'}}{|G_p| |G_q| |G_r| |G_s|}$$
$$= p\beta_{\ell - \ell'}.$$

Now, consider $p, q, p', q' \in \{1, \ldots, K\}$ such that $p \neq q$, $p' \neq q'$, $\ell^{lca}(G_p, G_q) = \ell + 1$ and $\ell^{lca}(G_{p'}, G_{q'}) = \ell$ for some $\ell \in \{0, 1, \ldots, L-1\}$. Thus, according to the planted model, one should merge $G_p, G_q$ before $G_{p'}, G_{q'}$. We verify that this is indeed the case under expectation since

$$\mathbf{E}[W(G_p, G_q)] - \mathbf{E}[W(G_{p'}, G_{q'})]$$
$$= \frac{1}{K(K-1)} \sum_{\substack{r,s=1 \\ r \neq s}}^{K} \mathbf{E}[\mathbb{W}_{\mathcal{Q}}(G_p, G_q \| G_r, G_s)] - \mathbf{E}[\mathbb{W}_{\mathcal{Q}}(G_{p'}, G_{q'} \| G_r, G_s)].$$
$$= \frac{1}{K(K-1)} \sum_{\substack{r,s=1 \\ r \neq s}}^{K} p\beta_{\ell+1-\ell^{lca}(G_r, G_s)} - p\beta_{\ell-\ell^{lca}(G_r, G_s)}$$
$$> p\Delta,$$

where the last step follows from arguments used in the proof of Theorem 3, which show that $\min_{\ell \in [-L, L-1]} \beta_{\ell+1} - \beta_\ell > \Delta$, where $\beta_\ell$ is defined in (5) and $\Delta$ is in the statement of the theorem.

Chaining of the above argument shows that $\mathbf{E}[W(G_p, G_q)] - \mathbf{E}[W(G_{p'}, G_{q'})] > p\Delta$ whenever $\ell^{lca}(G_p, G_q) > \ell^{lca}(G_{p'}, G_{q'})$. Under the assumptions stated in Theorem 4, we later prove that with probability $1 - \frac{\eta}{2}$,

$$\left| W(G, G') - \mathbf{E}[W(G, G')] \right| \leq \frac{p\Delta}{2} \tag{13}$$

for every pair of clusters $G, G'$ formed during the agglomerative steps of the algorithm starting from the given pure clusters of size in the range $[m, 2m]$. Based on (13) and the above argument, it is evident that $W(G_p, G_q) > W(G_{p'}, G_{q'})$ whenever $\ell^{lca}(G_p, G_q) > \ell^{lca}(G_{p'}, G_{q'})$ and, in particular, the cluster pair that achieves the maximum at any stage of iteration must be merged at the earliest according to the planted hierarchy. This guarantees exact recovery of the planted hierarchy by the algorithm.

We now prove (13). For this, we first state a concentration inequality that we prove later. Let $G_1, G_2, G_3, G_4$ be four clusters, each of size in the range $[m, 2m]$, such that $G_1, G_2$ are disjoint and so are $G_3, G_4$. Then

$$\mathbf{P}\left( \left| \mathbb{W}_{\mathcal{Q}}(G_1, G_2 \| G_3, G_4) - \mathbf{E}[\mathbb{W}_{\mathcal{Q}}(G_1, G_2 \| G_3, G_4)] \right| > \frac{p\Delta}{2} \right) \leq 2 \exp\left( 2\ln N - \frac{p\Delta^2 m^2}{C'} \right) \tag{14}$$

for some absolute constant $C' > 0$. We wish to use (14) to argue that with probability $1 - \frac{\eta}{2}$, all clusters in the initial partition (assumed in the theorem) satisfy the condition $|\mathbb{W}_{\mathcal{Q}}(G_1, G_2 \| G_3, G_4) - \mathbf{E}[\mathbb{W}_{\mathcal{Q}}(G_1, G_2 \| G_3, G_4)]| \leq \frac{p\Delta}{2}$. Note that we do not know how the initial partition is achieved, but we can ensure that

$$\mathbf{P}\Big( \exists G_1, G_2, G_3, G_4 : m \leq |G_1|, |G_2|, |G_3|, |G_4| \leq 2m,$$

$$|\mathbb{W}_{\mathcal{Q}}(G_1, G_2 \| G_3, G_4) - \mathbf{E}[\mathbb{W}_{\mathcal{Q}}(G_1, G_2 \| G_3, G_4)]| > \frac{p\Delta}{2} \Big)$$

$$\leq \sum_{i_1, i_2, i_3, i_4 = m}^{2m} \binom{N}{i_1}\binom{N}{i_2}\binom{N}{i_3}\binom{N}{i_4} 2 \exp\left( 2\ln N - \frac{p\Delta^2 m^2}{C'} \right)$$

$$\leq 2m^4 \left( \frac{eN}{m} \right)^{8m} \exp\left( 2\ln N - \frac{p\Delta^2 m^2}{C'} \right).$$

$$\leq C'' \exp\left( 9m\ln N - \frac{p\Delta^2 m^2}{C'} \right),$$

where $C'' > 0$ is an absolute constant such that $\sup_{m \geq 1} 2m^4 (\frac{e}{m})^{2m} < C''$. The above probability is bounded by $\frac{\eta}{2}$ for $p > \frac{C}{m\Delta^2}\left( \ln N \vee \frac{1}{m}\ln\left(\frac{1}{\eta}\right) \right)$ for some constant $C > 0$. Thus, with probability $1 - \frac{\eta}{2}$, we know that for every tuple of four clusters, obtained at initialization, $\mathbb{W}_{\mathcal{Q}}$ deviates from its mean by at most $\frac{p\Delta}{2}$. In fact, the same deviation also holds when we merge some of these clusters. For instance, let $G_1, G_1', G_2, G_3, G_4$ be part of a partition at some stage and suppose $G_1, G_1'$ are merged. Then

$$\mathbb{W}_{\mathcal{Q}}(G_1 \cup G_1', G_2 \| G_3, G_4) = \frac{|G_1|}{|G_1| + |G_1'|} \mathbb{W}_{\mathcal{Q}}(G_1, G_2 \| G_3, G_4)$$

$$+ \frac{|G_1'|}{|G_1| + |G_1'|} \mathbb{W}_{\mathcal{Q}}(G_1', G_2 \| G_3, G_4),$$

which is a convex combination of $\mathbb{W}_{\mathcal{Q}}$ computed at the previous stage. Hence, if each of them deviates from its mean by at most $\frac{p\Delta}{2}$, then the convex combination after merging also deviates from its mean by at most $\frac{p\Delta}{2}$. The same also holds for other instances of merging throughout the hierarchy, which shows that with probability $1 - \frac{\eta}{2}$, at any stage of agglomeration, we have $|\mathbb{W}_{\mathcal{Q}}(G_p, G_q \| G_r, G_s) - \mathbf{E}[\mathbb{W}_{\mathcal{Q}}(G_p, G_q \| G_r, G_s)]| < \frac{p\Delta}{2}$ for any tuple of four clusters in the partition. Now, observe that $W(G_p, G_q)$ is an average of several $\mathbb{W}_{\mathcal{Q}}$, and so, (13) holds.

We complete the proof of Theorem 4 by proving the concentration inequality in (14). Since $w_{ij} = w_{kl}$ occurs with zero probability for any $i, j, k, l (i \neq j, k \neq l)$, we may write

$$|\mathbb{W}_{\mathcal{Q}}(G_1, G_2 \| G_3, G_4) - \mathbf{E}[\mathbb{W}_{\mathcal{Q}}(G_1, G_2 \| G_3, G_4)]|$$

$$= \frac{2}{|G_1|\,|G_2|\,|G_3|\,|G_4|} \left| \sum_{x_i \in G_1} \sum_{x_j \in G_2} \sum_{x_k \in G_3} \sum_{x_l \in G_4} \left( \xi_{ijkl} \mathbb{I}_{(w_{ij} > w_{kl})} - p\mathbf{P}(w_{ij} > w_{kl}) \right) \right|,$$

where $\xi_{ijkl}$ is the indicator of observing the comparison between $(i, j)$ and $(k, l)$. Note that each term in the summation is a centred random variable in the range $[-1, 1]$, and has variance bounded by $p$. Let us denote each of them by $B_{ijkl}$, and observe that they have dependencies among themselves. We use the concentration technique of Janson and Ruciński (2002). Consider the dependency graph for these variables, which is a graph on $s = |G_1||G_2||G_3||G_4|$ vertices and two vertices are adjacent if they are dependent. Some of the vertices have degree $|G_1||G_2| - 1$ (dependent with other variables with same $k, l$), while other vertices have degree $|G_3||G_4| - 1$. Let us denote the maximum degree by $d$. One can find an equitable colouring for such a graph using $(d+1)$ colours, where equitable denotes that all colour classes are of nearly equal sizes $\lfloor \frac{s}{d+1} \rfloor$ or $\lceil \frac{s}{d+1} \rceil$. Denoting the colour classes by $\mathcal{C}_1, \ldots, \mathcal{C}_{d+1}$, we can bound the probability using the union bound and Bernstein's inequality as

$$\mathbf{P}\left( |\mathbb{W}_{\mathcal{Q}}(G_1, G_2 \| G_3, G_4) - \mathbf{E}[\mathbb{W}_{\mathcal{Q}}(G_1, G_2 \| G_3, G_4)]| > \frac{p\Delta}{2} \right)$$

$$= \mathbf{P}\left( \left| \sum_{i,j,k,l} B_{ijkl} \right| > \frac{sp\Delta}{4} \right)$$

$$\leq \sum_{\ell=1}^{d+1} \mathbf{P}\left( \left| \sum_{(i,j,k,l) \in \mathcal{C}_\ell} B_{ijkl} \right| > \frac{sp\Delta}{4(d+1)} \right)$$

$$\leq \sum_{\ell=1}^{d+1} 2\exp\left( -\frac{\frac{s^2 p^2 \Delta^4}{16(d+1)^2}}{2p|\mathcal{C}_\ell| + \frac{2}{3}\frac{sp\Delta}{4(d+1)}} \right).$$

The bound in (14) follows by first noting that $|\mathcal{C}_\ell| \leq \frac{2s}{d+1}$, and then using the fact $\frac{s}{d+1} \geq \min\{|G_1||G_2|, |G_3||G_4|\} \geq m^2$. For the outer summation, we simply use $(d+1) \leq N^2$ to obtain the bound in (14).

To verify the claim for fixed $L$ and $\frac{\delta}{\sigma}$, we note that, in this case, $\Delta$ is constant and $N_0 = \Omega(N)$. Hence, using $p = \frac{c \ln N}{m}$ for a large enough constant $c$ immediately leads to the exact recovery guarantee and the number of passive comparisons. $\qquad\square$

## B  Details on the experiments

In this section we present some details on the experiments that are not included in the main paper along with some additional plots and discussions.

### B.1  Planted Hierarchical Model

**Evaluation function.** As a measure of performance we report the Averaged Adjusted Rand Index (AARI) between the ground truth hierarchy $\mathcal{C}$ and the hierarchies $\mathcal{C}'$ learned by the different methods. Let $\mathcal{C}^\ell$ and $\mathcal{C}'^\ell$ be the partitions of $\mathcal{X}$ at level $\ell$ of the hierarchies, then:

$$\text{AARI}(\mathcal{C}, \mathcal{C}') = \frac{1}{L} \sum_{\ell \in \{1, \ldots, L\}} \text{ARI}\left( \mathcal{C}^\ell, \mathcal{C}'^\ell \right)$$

where ARI is the Adjusted Rand Index (Hubert and Arabie, 1985), a widely used measure to compare partitions. We use the average across the different levels $\mathcal{C}^\ell$ and $\mathcal{C}'^\ell$ to take into account the hierarchical structure. The AARI takes values in the interval $[0, 1]$ and the higher the value the more similar the hierarchies are. $\text{AARI}(\mathcal{C}, \mathcal{C}') = 1$ implies that the two hierarchies are identical. For all the experiments we report the mean and the standard deviation of 10 repetitions.

Figure S.1: AARI of the proposed methods (higher is better) on data obtained from the planted hierarchical model with $\mu = 0.8$, $\sigma = 0.1$, $L = 3$, $N_0 = 30$ and different sampling proportions $p$. Best viewed in color.

**Results.** In Figure S.1 we present supplementary results for the planted hierarchical model, that is with $p \in \{0.01, 0.02, \ldots, 0.1, 1\}$. Firstly, similar to the theory, SL can hardly recover the planted hierarchy, even for large values of $\frac{\delta}{\sigma}$. CL performs better than SL, which is not evident from the theory. This suggests that a better sufficient condition might be possible for CL. We observe that 4K–AL, 4K–AL–act, and, 4–AL are able to exactly recover the true hierarchy for smaller signal-to-noise ratio and their performances do not degrade much when the number of sampled comparisons is reduced. Finally, as expected, the best methods are 4–AL–I3 and 4–AL–I5. They use large initial clusters but recover the true hierarchy even for very small values of $\frac{\delta}{\sigma}$.

## B.2 Standard Clustering Datasets

**Data.** We provide some details on the datasets used in the paper. We evaluate the different approaches on 3 different datasets commonly used in hierarchical clustering: Zoo, Glass and 20news (Heller and Ghahramani, 2005; Vikram and Dasgupta, 2016). The Zoo dataset is composed of 100 animals with 16 features (it originally contains 101 animals but we chose to remove the 'girl' entry since we feel that it does not fit in a Zoo dataset). The Glass dataset has 9 features for 214 examples. The 20news dataset is composed of 11314 news articles. Following Vikram and Dasgupta (2016) we pre-processed the 20news dataset using a bag of words approach followed by PCA to retain 100 relevant features. We randomly sampled 200 examples for hierarchical clustering. To fit the comparison-based setting we generate the quadruplet comparisons using the cosine similarity:

$$w_{ij} = \frac{\langle \mathbf{x}_i, \mathbf{x}_j \rangle}{\|\mathbf{x}_i\| \, \|\mathbf{x}_i\|}$$

where $\mathbf{x}_i$ and $\mathbf{x}_j$ are the representations of objects $x_i$ and $x_j$ and $\langle \cdot, \cdot \rangle$ is the dot product. Since it is not realistic to assume that all the comparisons are available, we use the procedure described in Section 2.3 in the main paper to passively obtain a proportion $p \in \{0.01, 0.02, \ldots, 0.1\}$ of all the quadruplets. Note that tSTE-AL and FORTE-AL are based on ordinal embedding methods that use triplet comparisons of the form "object $i$ is more similar to object $j$ than to object $k$", that is $w_{ij} > w_{ik}$, rather than quadruplet comparisons. Nevertheless, we can use the same procedure than for the quadruplets to generate the same proportion of triplets that we can use in tSTE and FORTE. To the best of our knowledge, there does not exist ordinal embedding methods based only on quadruplet comparisons.

**Evaluation function.** Contrary to the planted hierarchical model we do not have access to a ground-truth hierarchy and thus we cannot use the AARI measure to evaluate the performance of the methods. Instead we use the recently proposed Dasgupta's cost (Dasgupta, 2016) that has been specifically designed to evaluate hierarchical clustering methods. Given a base similarity measure $w$, the cost of a hierarchy $\mathcal{C}$ is

$$\mathrm{cost}(\mathcal{C}, w) = \sum_{x, x_j \in \mathcal{X}} w_{ij} \left| \mathcal{C}^{lca}(x_i, x_j) \right|$$

where $w_{ij}$ is the similarity between $x_i$ and $x_j$ and $\mathcal{C}^{lca}(x_i, x_j)$ is the smallest cluster containing both $x_i$ and $x_j$ in the hierarchy. The idea of this cost is that similar objects that are merged higher in the hierarchy should be penalized. Hence, a lower cost indicates a better hierarchy. A low cost is achieved if similar objects (high $w_{ij}$) are merged towards the bottom of the tree (small $\mathcal{C}^{lca}(x_i, x_j)$), and vice-versa. Hence, a lower value of the cost indicates a better hierarchy. For all the experiments we report the mean and the standard deviation of 10 repetitions.

**Results.** In Figures S.2, S.3, and S.4 we present supplementary results for the standard clustering datasets. We note that the proportion of comparisons does not have a large impact as the results are, on average, stable across all regimes. Our methods are either comparable or better than the embedding-based ones. Our methods do not need to first embed the examples and thus do not impose a strong Euclidean structure on the data. The impact of this structure is more or less pronounced depending on the dataset. Furthermore, the performance of tSTE-AL and FORTE-AL depends on the embedding dimension that should be carefully chosen. For example, on Zoo, the performance of tSTE drops with increasing dimension. Similarly, on Glass, FORTE seems to perform slightly better for larger dimensions. Unfortunately, in clustering, tuning parameters can be difficult as there is no ground-truth.

## B.3 Comparison-based datasets

The Car dataset (Kleindessner and von Luxburg, 2017) is composed of 60 different type of cars and 6056 ordinal comparisons, collected via crowd-sourcing, of the form *Which car is most central among the three $x_i$, $x_j$ and $x_k$?*. These statements translate easily to the triplet setting: if $x_i$ is most central in the set of three then we recover two triplets $(j, i, k)$ and $(k, i, j)$. Then triplet comparisons further translate into quadruplet comparisons by noticing that the triplet $(i, j, k)$ corresponds to the quadruplet $(i, j, i, k)$. Overall we obtained 12112 comparisons that we used to learn a hierarchy among the cars.

Figure S.2: Dasgupta's score of the different methods on the Zoo dataset with increasing embedding dimensions for FORTE–AL and tSTE–AL. Best viewed in color.

Figure S.3: Dasgupta's score of the different methods on the Glass dataset with increasing embedding dimensions for FORTE–AL and tSTE–AL. Best viewed in color.

**(a) Dimension = 2**

**(b) Dimension = 3**

**(c) Dimension = 4**

**(d) Dimension = 5**

Figure S.4: Dasgupta's score of the different methods on the 20news dataset with increasing embedding dimensions for FORTE–AL and tSTE–AL. Best viewed in color.

The hierarchies obtained by 4K–AL, 4–AL, FORTE–AL and tSTE–AL are attached to this supplementary as png files. The names of the files are respectively cars.4K–AL.png, cars.4–AL.png, cars.FORTE–AL.embedding_dimension.png and cars.tSTE–AL.embedding_dimension.png.