[Reviews · NeurIPS 2019]

Reviewer 1



In this work the authors study hierarchical clustering under quadruplet comparison framework. The authors show that single and complete linkages are inherently comparison based and propose two variants of average linkage clustering exploiting quadruplet comparison. Exact hierarchy recovery guarantee is provided under planted hierarchical partition model and empirical evaluation is provided. 1. The meaning of the variables \mu, \delta etc are hard to interpret from the description. They have been nicely summarized (and explained) in the appendix A.1. May be placing these descriptions in the main paper will improve readability. 2. The drawback of theorem 2 is that size of pure cluster N_0 to be of the order \sqrt(N). For large N, this is a bit restrictive, is it possible to improve N_0 to be logarithmic in N or perhaps \theta(1)? Otherwise, is it possible to prove that no recovery is possible unless N_0 is at lest \sqrt(N)? 3. The main difference between active and passive sampling is that in case of passive sampling comparison queries are asked by flipping a Bernoulli random variable out of all O(N^4) quadruplets. Clearly many of such quadruplets may be uninformative thus requiring more comparison queries. In case of active sampling, first a tuple (i_0, j_0) is fixed uniformly at random and the the rest of the queries are asked with respect to this chosen tuple. Clearly, uniformly choosing a single tuple will have high variance and may be using more than one such uniformly chosen sample will reduce variance as was suggested for empirical evaluation (equation 2). Is it possible to quantify the reduction in variance and come up with the size of the set \mathcal{R} in equation 2? In particular, can it be the case that a properly chosen \mathcal{R} will result in better than O(N log N) comparison complexity? 4. In equation 5, K seems to be the number of clusters. But K will vary at different level of the dendrogram? Is that interpretation correct? 5. Interpretation of Theorem 4 in lines 256-259 is confusing. If the number of passive comparisons is O(N^4 log N /m) as given in line 251 and m=\Omega( log N), then number of passive comparison required is O(N^4) that is all quadruplets needs to be ovserved, in line 256-257 it is claimed otherwise. Similarly Only when m=\Omega (N) (in line 257 should it be \Omega (N) instead of \Omega(N_0)?) required number of comparison queries is O(N^3 log N). But then pure cluster of size m=\Omega(N) is kind of useless is not it (each pure cluster is just too big)? 6. In plants hierarchical model experiment size of pure cluster (N_0) =30. in 4-AL-I3, initial cluster size is 3, how is initial cluster different from pure cluster? In case of 4-AL, is the size of pure cluster (N_0) still 30? In this case number of levels L=3. Now in case of 4-AL-I3, where initial cluster size is 3, the level has to be greater than 3 if N is fixed in both cases. Can you please clarify this issue? 7. Since there is no common element in quadruplet comparison as opposed to triplet comparison Is it possible that quadruplet comparison may have inherent noise. For example, in the crowdsourcing setting, asking the question “Is similarity between dolphins and whales larger than similarity between tigers and panthers” might result in different answers. In absence of an omnipotent oracle, how to tackle such inherent noise/bias towards perfect reconstruction?

Reviewer 2



- Such a query setup may be done in the following mode: * Active queries: Here the clustering algorithm generates comparison queries during its execution and once these queries are answered, it continues its execution and possibly generate more queries. Note that this may involve a lot of time since the clustering algorithm’s time depends on the queries being answered. * Passive queries: The clustering algorithm decides all the queries before its execution. - This paper specifically discusses the effectiveness of quadruplet queries in the context of the planted partition model. Single linkage, complete linkage, and average linkage are popular algorithms in the classical hierarchical clustering framework where pairwise distances are known. These involve finding the maximum/minimum/average distance between a pair of points in two clusters. In the new model of quadruplet queries finding the max/min are still possible using the queries. So, it is still possible to implement the single linkage and complete linkage in the new setting. However, it is not immediately clear how to implement the average linkage algorithm in the comparison model. They suggest a method to define distance between points/clusters using the comparison queries and this gives an average-linkage based algorithm for agglomerative hierarchical clustering. The effectiveness of this algorithm in the context of planted partition model is discussed and theoretical guarantees are given. Experimental results are also given for real datasets where comparison is made with respect to the hierarchical clustering score given by Dasgupta. - It would be good to see a theorem for average linkage similar to theorem 1 that is for single linkage and complete linkage. - It seems that the comparison based algorithm is giving good Dasgupta score for some real datasets. Can some theoretical guarantees be obtained? - Can lower bounds on the number of queries be given in the comparison setting? Overall, this paper starts a very nice discussion. It would be even stronger if more questions in the context are explored.

Reviewer 3



This paper is original and solves an important and interesting problem. It is also very well written and I loved reading it. However I have the following questions: 1) In Theorem 1, exact conditions on \Delta/ \sigma is provided which is improved in the subsequent theorems. However, the necessary conditions on \Delta/ \sigma are not provided in the subsequent theorems explicitly. Is it possible to provide a table/plot for comparison? 2) Can you provide an intuition behind why average linkage performs better? Why not try something else besides these popular linkage algorithms such as median linkage? Median will definitely be more robust to outliers. Is it known that average linkage is the best someone can do in these settings? 3) Are each comparisons sampled multiple times in the algorithm to smooth the noise? Note that asking the same query multiple times might not be good in practice although it will work in theory.

[Author Response · NeurIPS 2019]

We thank the reviewers for their detailed comments. Few questions require a more detailed analysis (lower bounds). We
respond to all the comments, but postpone some further analysis to a longer version of the paper that is in preparation.

**Reviewer 1: 1. Comparison with Emamjomeh-Zadeh and Kempe [EK18].** A table may confuse the reader since
their setup differs from ours, but we will include the following discussion. [EK18] considers noise in the comparisons
rather than in the similarities, and studies triplets rather than quadruplets. They show that: (i) $O(N \ln N)$ active
comparisons are sufficient (we match this in Theorem 2), and (ii) $\Omega(N^3)$ passive triplets are necessary (for quadruplets,
their proof leads to $\Omega(N^4)$ passive quadruplets are necessary; we obtain better guarantees, but under a planted model).

**2. Improving $N_0 = \Omega(\sqrt{N})$ condition.** Note that $N_0 = \Omega(\sqrt{N})$ implies there are $O(\sqrt{N})$ pure clusters. In the
standard stochastic block model literature, there are no known poly-time algorithm that can exactly recover $\omega(\sqrt{N})$
planted clusters (see Figure 1 of Chen & Xu, arXiv:1402.1267). Hence, this condition could be optimal for exact
recovery under a planted model in the sense that it is necessary for any poly-time algorithm.

**3. Can larger $\mathcal{R}$ reduce the comparison complexity?** The analysis is possible, but the improvement would be at
most $\ln N$ since the active comparison complexity for exact recovery is $\Omega(N)$ to have at least one comparison per item.

**4. Does $K$ vary at different levels?** Yes, after every merge the number of clusters $K$ is reduced by one.

**5. Interpretation of Theorem 4.** (i) The condition $m = \Omega(\ln N)$ should have been $\omega(\ln N)$, which will be corrected.
(ii) $m = \Omega(N)$ can be obtained through flat clustering using kernel (3). Large initial clusters are also needed for
recovering a planted hierarchy using average linkage with known similarities [Cohen-Addad et al. 2018; Theorem 5.8].

**6. How is initial cluster different from pure cluster in 4–AL–I3?** The term "pure clusters" is probably confusing.
The clusters at the bottom of the planted hierarchy are "pure clusters" of size $N_0 = 30$. Then, following Theorem 4,
4–AL has to be initialized with small "pure clusters" of size $m$ (pure means that they are sub-clusters of one of the
bottom clusters in the planted hierarchy). For 4–AL–I3, we set $m = 3$. For 4–AL, we set $m = 1$.

**7. Inherent noise in quadruplet queries.** To deal with noise in the quadruplets (that is quadruplets that are flipped
compared to an omnipotent oracle), one may assume that each query is independently incorrect with some probability,
as in crowdsourcing. Then, it is possible to show that the proposed algorithms can recover the hierarchy under similar
sufficient conditions. We will address this issue in the longer version of the paper. In the current submission, we focus
on the problem of noise in the similarities as it brings more novelty to the field ([EK18] studies flipped comparsions).

**Reviewer 3: 1. Theorem for average linkage similar to Theorem 1.** This comment is not clear to us. If it is about
necessary $\frac{\delta}{\sigma}$, see our answer to Comment-1 of Reviewer-4. If it asks for lower bound, we respond to your Comment-3.

**2. Theoretical guarantees for Dasgupta's score.** Under a planted model, exact recovery corresponds to achieving a
$(1 + o(1))$-approximation of the optimal Dasgupta's score. Obtaining a worst-case guarantee for arbitrary data would be
more difficult in our comparison setting. Indeed, Dasgupta's score, and the associated theoretical results, all heavily rely
on the fact that true values of the similarities can be accessed. Following this, an interesting future research direction
would be to derive an ordinal variant of Dasgupta's score based on comparisons.

**3. Can lower bounds on the number of queries be given?** Yes, but we do not have a complete picture yet. We only
provide insights here. In the active setting, it has to be at least $\Omega(N)$ to observe at least one comparison per item. We
nearly match this bound with the $O(N \ln N)$ upper bound in Theorem 2 (in a special case). In the passive setting, there
are possibly two cases depending on whether the SNR is constant or grows with $N$, which we will explore in the future.

**Reviewer 4: 1. Necessary conditions for $\frac{\delta}{\sigma}$.** The necessity of $\frac{\delta}{\sigma} = \Omega(\sqrt{\ln N})$ is established in Theorem 1 to show
that single linkage only recovers the hierarchy under a very restrictive scenario where the SNR grows with $N$. This
does not happen in the subsequent theorems, which all include the special case of $\frac{\delta}{\sigma} = O(1)$. We feel this is reasonable,
and hence, state the sufficient conditions in terms of $N_0$. Assuming $\frac{\delta}{\sigma} = O(1)$, the sufficient condition $N_0 = \Omega(\sqrt{N})$,
stated in Theorems 2-3, is also necessary (see our answer to Comment-2 of Reviewer-1).

**2. Why average linkage performs better? Why not other linkages?** Average linkage is generally better since
averaging tends to reduce the noise. For 4K–AL, there are two levels of averaging – the kernels in (1-3) are sums, and
we use average linkage. Similarly, for 4–AL, the averaging in (4) is crucial to counteract the noise. In this paper, we
focus on the most popular linkage methods but it would also be interesting to further study other linkages. In particular,
the kernels in Equations (1-3) can be combined with similarity-based linkages. Developing a median linkage variant of
4–AL would, however, require a more extensive study.

**3. Is each comparison sampled multiple times by the algorithm?** No, in the current theoretical study, we assume
that each comparison is observed exactly once (in both the active and the passive case). In the experiments, we only
query the comparisons once in the active case. In the passive case (in particular on the real dataset), it might happen that
the same comparison is given several times. In this case, we use a majority vote (that is, use the more frequent answer).

[Meta-Review · NeurIPS 2019]

The authors have proposed two variants of average linkage hierarchical clustering using quadruplet comparison framework. Theoretical results of hierarchy recovery is established under a suitable model. The reviewers are in agreement that the results are new and important. The authors should incorporate the suggestions made by the reviewers to further strengthen the paper.